# Predicting memory from the network structure of naturalistic events

Hongmi Lee 🄸 1✉ & Janice Chen 🄸 1

When we remember events, we often do not only recall individual events, but also the connections between them. However, extant research has focused on how humans segment and remember discrete events from continuous input, with far less attention given to how the structure of connections between events impacts memory. Here we conduct a functional magnetic resonance imaging study in which participants watch and recall a series of realistic audiovisual narratives. By transforming narratives into networks of events, we demonstrate that more central events—those with stronger semantic or causal connections to other events—are better remembered. During encoding, central events evoke larger hippocampal event boundary responses associated with memory formation. During recall, high centrality is associated with stronger activation in cortical areas involved in episodic recollection, and more similar neural representations across individuals. Together, these results suggest that when humans encode and retrieve complex real-world experiences, the reliability and accessibility of memory representations is shaped by their location within a network of events.

1 Department of Psychological and Brain Sciences, Johns Hopkins University, Baltimore 21218 MD, USA. ✉email: hongmi.lee@jhu.edu

   

Remembering the experiences of our lives requires collecting and connecting the pieces and reasons for what transpired. When we tell each other about the minutes and hours leading up to this moment, the tale will be composed of a string of time periods, "events"[1,2], distinguished by properties such as their locale or mood, and by our companions or goals at the time. Traditional experimental memory paradigms[3,4] rely on isolated stimuli in which meaningful connections between memoranda across time are removed via trial randomization. Yet in reality, each event exists within, and is to some extent defined by, a dense network of connections across time. These connections come in multiple forms: different timepoints could share properties to greater or lesser degrees, and actions earlier may have consequences later. When remembering and retelling, we often need to recapitulate not only the most important individual events, but also the overall structure of the experience, i.e., the pattern of connections across time[5,6]. Thus, it is important to understand in what ways the web of interrelations between events contributes to our memories of those experiences.

In order to test how inter-event structure relates to later memory, experimenters must use study material which contains inter-event structure. Recently, researchers have sought to incorporate the complex, multi-event nature of real-world input into laboratory experiments by using auditory and/or visual narratives[7,8]. Since narratives are temporally continuous, a major question in the literature has been how the human brain identifies and remembers discrete events from continuous experiences[2,9,10]. As input arrives from the world, the perceiver constructs a mental model of the situation, which consists of agents, objects, spatio-temporal contexts, and the relations between these components[11]. Changes in the ongoing situation trigger the registration of the just-concluded event into long-term memory, evoking transient responses in the hippocampus and its cortical partners[12,13]. The boundaries between events are also associated with shifts in neural activation patterns in higher associative areas in the default mode network (DMN[14])[15]. DMN activity patterns specific to individual events are thought to represent situation models[16], and are reinstated during narrated memory recall[17,18]. However, these studies focus on how each event is segmented from its temporally adjacent neighbors. How do the myriad connections between events, both temporally proximal and distal, impact the cognitive and neural underpinnings of naturalistic memory?

Inter-event connections could benefit both memory encoding and retrieval. At encoding, events with strong connections to numerous other events might be frequently reactivated by these links to form robust and integrated representations[19,20]. At retrieval, events with many connections might be more likely to be cued by other events, enhancing their accessibility. These enhancing effects of inter-event connections on memory have been demonstrated in the reading comprehension literature, which focused on casual relations in relatively short and carefully designed text passages[21,22]. For example, statements that form causal chains are better remembered than isolated statements, and memory accuracy for a statement increases with the number of causal connections that it has[22,23]. Causal connectivity between statements also predicts how important readers will deem a given statement to be, and what they will judge to be the gist of the narrative[22,24]. The current study aims to examine the mnemonic benefits of inter-event connections in light of the burgeoning cognitive neuroscience of memories for events. Using previously unavailable neuroimaging approaches, we investigate the effect of inter-event structure on brain functions supporting the encoding and retrieval of event representations. In addition to testing the influence of causal relations, we take advantage of natural language processing techniques which allow effortless quantification of semantic similarity between text descriptions of complex events[25–27]. These non-causal (semantic) relations, based on shared meaning and overlapping components between events, may constitute a previously underexplored pathway through which inter-event connections enhance memory.

Here, we propose that when people view and recall realistic, continuous audiovisual stimuli (e.g., movies), events with stronger and more numerous semantic or causal connections to other events will be better remembered, with concomitant hippocampal and DMN activity reflecting enhanced encoding and retrieval-related processing for these events. We conducted a functional magnetic resonance imaging (fMRI) study in which participants watched a series of movies and then verbally recounted the movie plots. To quantify and assess the semantic relationship between events within a movie, we employed an approach scalable and easily generalizable to different types of narratives (Fig. 1). In this method, each narrative is transformed into a network of interconnected events based on semantic similarity measured from sentence embedding distances (the semantic narrative network). We then calculate semantic centrality for each event as the node degree, a graph metric which quantifies the number and strength of connections that a node (event) has to other nodes in the network. Behavioral results revealed that events with higher semantic centrality are more likely to be recalled, without showing primacy and recency effects typical in traditional random list memory experiments[3,28]. High centrality events are also associated with the neural signatures of stronger and more accurate recall: greater activation and more consistent neural patterns across individuals in the DMN areas including the posterior medial cortex (PMC). The hippocampus shows higher activation following the offset of high centrality events, suggesting that stronger hippocampus-mediated encoding contributes to the high centrality advantages. In parallel, we created a causal narrative network for each movie based on causal relations between events defined by human judgments. Causal centrality of events, again defined as node degree in the network, predicts memory success and neural responses in a similar way to semantic centrality, but also makes an independent contribution to each. Overall, our findings demonstrate that memories for events are shaped by their location within a narrative network, highlighting the importance of considering inter-event structure when studying the cognitive and neural mechanisms of complex and continuous real-world memory.

## Results

**Behavioral characteristics of unguided narrative recall.** We first examined the behavioral characteristics of free spoken narrative recall. Participants watched a series of short movies with unique narratives (Supplementary Table 1) and then verbally recalled the movie plots while undergoing functional MRI. Participants were instructed to describe what they remembered from the movies in their own words in as much detail as they could, regardless of the order of presentation. No external cues or experimenter guidance were provided during recall.

Two example participants' recall behaviors are depicted in Fig. 2a. On average, participants recalled 9 out of the 10 movies (s.d. 1.2) and the recall lasted 32.4 min in total (s.d. 14.5 min). Each movie was divided into 10–35 events by an independent coder based on major shifts in the narrative (e.g., time, location, action). Participants on average recalled 77.6% of the events within each recalled movie (s.d. 11.2%). Movies tended to be recalled in the original presentation order (mean Spearman's $\rho$ between the presentation order and the recalled order = .52, s.d. across participants = .55; Fig. 2b, top panel). Although participants were not explicitly instructed to perform serial recall, events were typically recalled strictly in the order in which they occurred

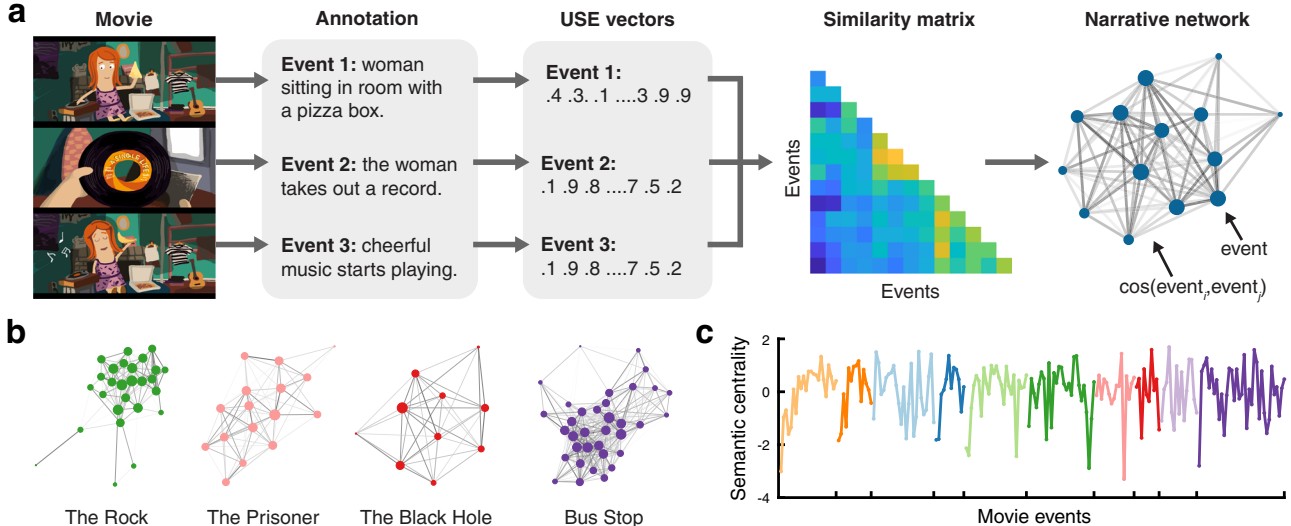

**Fig. 1 Semantic narrative networks. a** To create semantic narrative networks, each movie was split into events, and independent annotators provided text descriptions of the events. The text descriptions were transformed into sentence embedding vectors using Google's Universal Sentence Encoder (USE)[25]. Semantic similarity between events was computed as the cosine similarity between the USE vectors. A semantic narrative network was defined as a network whose nodes are movie events and the edge weights are the semantic similarity between the events. **b** Semantic narrative networks of four example movies used in the fMRI experiment. Edge weights were thresholded at cosine similarity = .6 for visualization purposes. Node size is proportional to centrality computed from unthresholded networks. Edge thickness is proportional to edge weights. **c** Semantic centrality (normalized degree) for individual movie events of the 10 movies used in the fMRI experiment. Different colors denote different movies. Source data are provided as a Source Data file. For the semantic similarity matrices and narrative networks of all 10 movies, see Supplementary Fig. 3. Movie scene images in **a** were created by the author H. L. using Adobe Illustrator (adobe.com).

within each movie (mean $\rho$ = .97, s.d. = .03; Fig. 2c). Thus, recalling an event likely served as a strong cue for the following event which was often semantically/causally related.

Contrary to traditional random list memory experiments[3,28], we did not observe the classic primacy and recency effects on recall probability[3] either at the movie level or the event level. The proportion of participants who successfully recalled a movie was not higher for the first or last few movies compared to the movies presented in the middle of the list (Fig. 2b, bottom panel). Likewise, the recall probability of the first/last few events was not higher than that of the events presented in the middle, either within each movie or across all movies (Fig. 2d). Specifically, we did not find a significant difference between the mean recall probabilities of the first/middle/last three events of each movie ($F(2,18)$ = .78, $p$ = .47, $\eta^2$ = .05). These results suggest that memorability of a movie event was largely influenced by narrative properties beyond the serial position of events.

**Narrative network centrality predicts what people will remember later.** One important factor that may have affected the behavioral characteristics of movie event recall is the inter-event structure inherent in narratives. We quantified narrative structure by transforming each movie plot into a network of events (Fig. 1), in which the connections between events were determined by their similarity based on semantic contents. To measure semantic similarity between movie events, we first converted the text descriptions of the events, generated by independent annotators, into vectors of 512 numbers using Google's Universal Sentence Encoder (USE[25]). Consistent with a recent study[26], the trajectories of movie annotations in the high-dimensional vector space were highly consistent across annotators (Supplementary Fig. 1), demonstrating that the text embeddings captured the semantic gist despite the differences in specific words used to describe the events. Likewise, the USE vectors of recall transcripts were similar to those of movie annotations and were also similar across

participants (Supplementary Fig. 2). Semantic similarity between events was defined as the cosine similarity between their USE vectors.

Our main variable of interest reflecting the inter-event narrative structure was the centrality of individual events within a narrative network (Fig. 1c). An event's centrality was computed as its degree (i.e., the sum of the weights of all connections to the event) normalized within each movie. Thus, events with stronger (higher semantic similarity) and greater numbers of connections with other events had higher centrality. Critically, semantic centrality positively predicted subsequent event recall probability, measured as the proportion of participants who recalled each event ($r(202) = .20$, $p = .004$, 95% confidence interval (CI) = [.07, .33]; Fig. 3a). To further test the effect of semantic centrality in individual participants, we grouped events into high or low centrality conditions within each movie (i.e., events whose semantic centrality values are within the top/bottom 40%), and measured the proportion of successfully recalled events in each condition. The recall probability averaged across movies was higher for the high than for the low semantic centrality condition ($t(14)$ = 6.12, $p < .001$, Cohen's $d_z$ = 1.58, 95% CI of the difference = [.06, .12]; Fig. 3b).

We next demonstrated that inter-event semantic relations and causal relations made overlapping as well as unique contributions to narrative memory performance. Classic studies on story comprehension have reported that the number of causal connections with other events predicts the perceived importance and memorability of an event[22,24]. To test the effect of causal relations, we created the causal narrative networks of the movies (Supplementary Fig. 4) by having independent coders identify causally related events within each movie (see Supplementary Fig. 5 and Supplementary Methods for detailed descriptions of causality responses and the instructions given to the coders). The connection strength between a pair of events was defined as the proportion of coders who responded that the pair is causally related. The centrality (i.e., normalized degree) of each event was

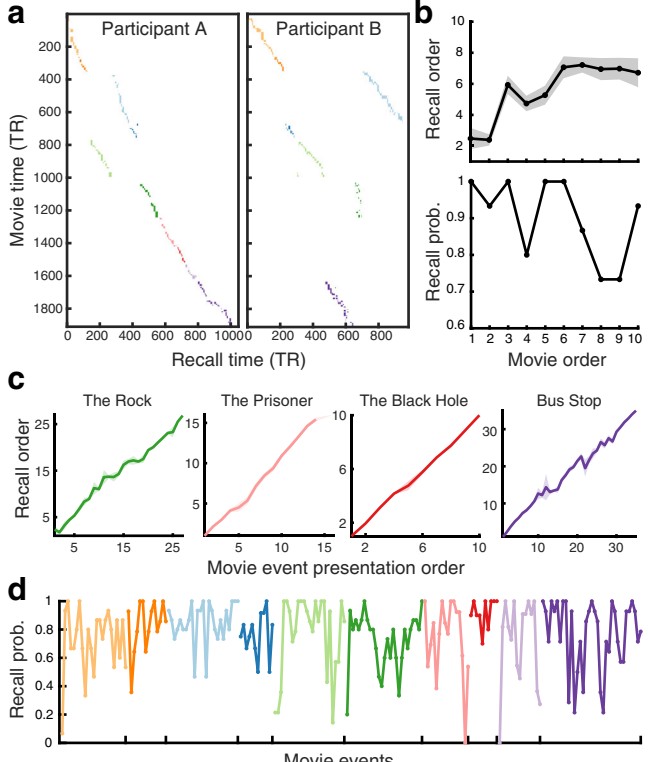

**Fig. 2 Unguided spoken narrative recall behavior. a** The duration and order of spoken recall for two example fMRI participants. Each colored rectangular dot represents a movie event. Different colors denote different movies. The *x* and *y* coordinates of a dot represent the temporal position of the event during recall and movie watching, respectively (TR = 1.5 s). The width and height of a dot represent the duration of the event during recall and movie watching, respectively. **b** Recall order (top) and recall probability (bottom) of the ten movies used in the fMRI experiment. **c** Recall order of individual movie events in four example movies. **d** Recall probability of individual movie events for the ten movies shown in different colors. In **b** and **c**, recall order was defined as the rank among recalled movies or events (i.e., 1 = recalled first, *N* = recalled last, where *N* is the total number of movies or events). Shaded areas indicate SEM across participants. In **b** and **d**, recall probability was calculated as the proportion of participants who recalled each movie or event. Source data for **b**–**d** are provided as a Source Data file.

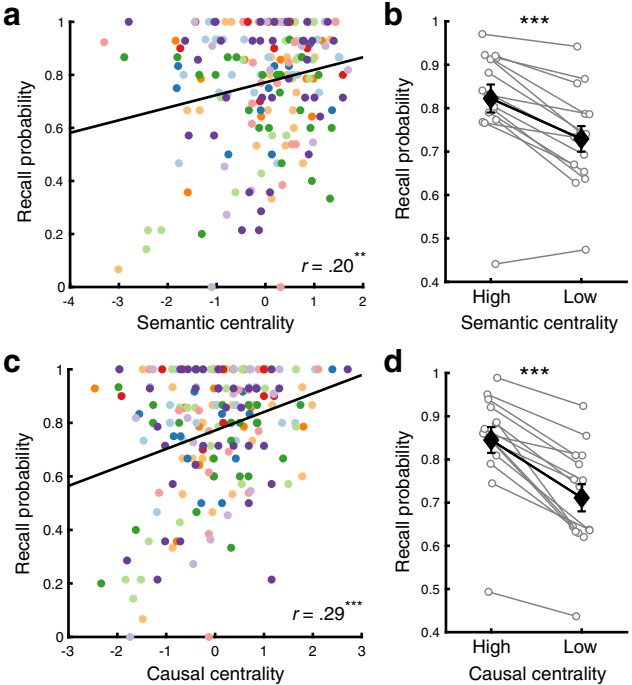

**Fig. 3 Effects of narrative centrality on recall performance. a** Correlation between semantic centrality and recall probability. **b** Recall probability for High (top 40%) vs. Low (bottom 40%) semantic centrality events defined within each movie (averaged across movies). **c**. Correlation between causal centrality and recall probability. **d**. Recall probability for High (top 40%) vs. Low (bottom 40%) causal centrality events defined within each movie (averaged across movies). In **a** and **c**, each dot represents an individual movie event. Different colors denote different movies. In **b** and **d**, white circles represent individual participants (*N* = 15). Black diamonds represent the mean across participants within each condition. Error bars show SEM across participants. Two-tailed paired t-tests indicated that both higher semantic (*p* = .00003) and causal centrality (*p* = .000001) were associated with higher recall probability. **\*\****p* < .01, **\*\*\****p* < .001. Source data are provided as a Source Data file.

then computed within each causal narrative network. Causal centrality was positively correlated with semantic centrality (*r*(202) = .28, *p* < .001, 95% CI = [.15, .41]) and recall probability (*r*(202) = .29, *p* < .001, 95% CI = [.16, .42]; Fig. 3c). Recall probability was also higher for high than for low causal centrality events within each participant (*t*(14) = 8.23, *p* < .001, Cohen's $d_z$ = 2.12, 95% CI of the difference = [.1, .17]; Fig. 3d), consistent with earlier studies[22,23]. Importantly, a mixed-effects logistic regression analysis revealed that semantic centrality explains successful event recall even after controlling for causal centrality ($\beta$ = .17, standard error (SE) = .05, $\chi^2$(1) = 12.24, *p* < .001) and vice versa ($\beta$ = .38, SE = .05, $\chi^2$(1) = 55.04, *p* < .001).

We conducted a preregistered online experiment (*N* = 393) and replicated the same behavioral characteristics of narrative recall using a new set of 10 short movies (Supplementary Fig. 7). Each participant watched one of the movies and then performed a free written recall of the movie plot. Consistent with the behavioral results from the fMRI experiment, semantic centrality ($\beta$ = .17, SE = .03, $\chi^2$(1) = 48.52, *p* < .001) and causal centrality ($\beta$ = .44, SE = .03, $\chi^2$(1) = 255.67, *p* < .001) each uniquely

predicted the successful recall of an event, without any clear evidence of serial position effects (i.e., no statistically significant difference between the mean recall probabilities of the first/middle/last three events of each movie, *F*(2,18) = .85, *p* = .44, $\eta^2$ = .04).

**High centrality events more strongly activate DMN during recall.** Narrative network centrality predicted successful memory recall of movie events. Does it also predict brain responses associated with movie watching and recall? We first identified brain regions whose activation scaled with the semantic centrality of events. In this and all following analyses, we excluded the first event of each movie from the movie watching data. This was to minimize the influence of transient changes in activation associated with the boundaries between narratives[12,29]. The movie boundary-related responses also disrupted event-specific neural patterns by creating similar activation patterns across all movies (Supplementary Fig. 8).

We performed a whole-brain general linear model (GLM) analysis designed to predict the mean activation of individual events with their semantic centrality. Group-level analysis of the participant-specific beta maps showed that, at a liberal threshold (uncorrected *p* < .001), higher semantic centrality of an event was associated with stronger activation in several regions including visual and auditory association cortices and

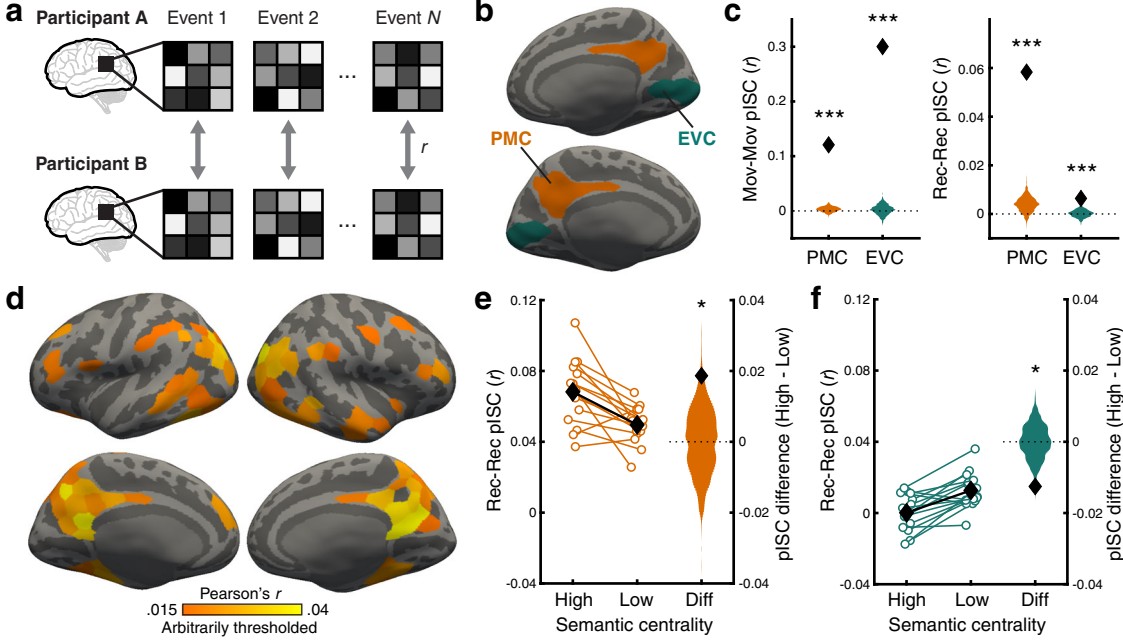

**Fig. 4 Event-specific intersubject pattern correlation. a** Intersubject pattern correlation (pISC) was computed for each movie event by correlating the event-specific activation pattern (averaged across times within the event) of a participant and that of each of the other participants. **b** Posterior medial cortex (PMC; orange) and early visual cortex (EVC; green) regions-of-interest visualized on the inflated surface of a template brain (medial view). **c** pISC in PMC and EVC during movie watching (left) and recall (right). Black diamonds show the mean pISC averaged across all participants and movie events. Orange and green histograms show the null distributions of the mean pISC in PMC and EVC, respectively. Statistical significance reflects difference from zero based on one-tailed randomization tests (all $p$s = .000999). **d** Whole-brain surface map of mean pISC during recall. pISC was computed for each of 400 parcels in a cortical atlas[33]. The pISC map was arbitrarily thresholded at $r$ = .015 for visualization purposes. pISC values in all visualized parcels were significantly greater than zero based on randomization tests (FDR-corrected $q$ < .05 across parcels). **e, f** pISC for High vs. Low semantic centrality events during recall and the difference (Diff) between the two conditions in PMC (**e**) and EVC (**f**). For High and Low semantic centrality conditions, white circles represent individual participants ($N$ = 15). Black diamonds represent the mean across participants within each condition. Error bars show SEM across participants. For the difference between High and Low conditions (Diff), black diamonds show the true participant average, and histograms show the null distribution of the mean difference. Statistical significance reflects difference from zero based on two-tailed randomization tests ($p$ = .037 and .012 in PMC and EVC, respectively). *$p$ < .05, ***$p$ < .001. Source data for **c**, **e**, and **f** are provided as a Source Data file.

precuneus during movie watching (Supplementary Fig. 9a). The involvement of sensory areas may reflect high-level perceptual differences between the high and low centrality events, although low-level visual and auditory features including luminance, contrast, and audio amplitude were not significantly modulated by semantic centrality (all $\chi^2(1)$s < 1.94, $p$s > .16). More importantly, during recall, events with higher semantic centrality more strongly activated default mode network (DMN) areas including the angular gyrus and PMC (Supplementary Fig. 9b). DMN areas have been strongly associated with episodic recollection[16,30]. We also observed higher activation during recall for high than low semantic centrality events in the bilateral hippocampus ($t(14)$ = 2.71, $p$ = .017, Cohen's $d_z$ = .7, 95% CI of the difference = [.01, .05]). These results are in accordance with the positive relationship between recall performance and semantic centrality, and may suggest that high centrality events were more strongly recollected with rich episodic details.

We used causal centrality as a regressor in the GLM analysis and again found greater activation in the same DMN areas for higher centrality events during recall (Supplementary Fig. 9d). In this and following fMRI analyses, the effects of causal centrality were generally comparable to those of semantic centrality, except that causal centrality effects were weaker in analyses involving intersubject similarity. Thus, we focus on the semantic centrality effects and report the causal centrality effects in Supplementary Fig. 10. We consider potential differences between semantic and causal centrality in the Discussion.

**Neural patterns in DMN reflect both event-specific representations and narrative network structure.** Prior studies have shown that narrative events are represented as distributed patterns of activation in DMN[17,18]. How does inter-event structure relate to the neural representations of events during movie watching and recall? To answer this question, we performed an intersubject pattern correlation (pISC) analysis[17]. Within a brain region, event-specific pISC was computed as the mean spatial similarity (i.e., Pearson correlation) between a participant's activation pattern of a given event and each of the other participants' activation patterns of the same event (Fig. 4a). By measuring neural signals shared across participants, the intersubject correlation method was expected to reduce the influence of task-unrelated idiosyncratic noise[31,32].

We first created whole-brain pISC maps to identify brain regions that showed robust event representations shared across participants. For each cortical parcel of an atlas[33], we computed the mean pISC averaged across events and participants. We then performed a nonparametric randomization test to determine whether the mean pISC was significantly different from a null distribution generated by randomly shuffling event labels across all movies. Replicating our prior study[17], positive pISC was observed in widespread cortical regions during both movie watching and recall (FDR corrected $q$ < .05). During movie watching (Supplementary Fig. 11a), the strongest pISC was found in sensory cortices, as all participants processed the same audiovisual stimuli. During recall (Fig. 4d), DMN areas, especially PMC, showed the strongest pISC, consistent with the view that

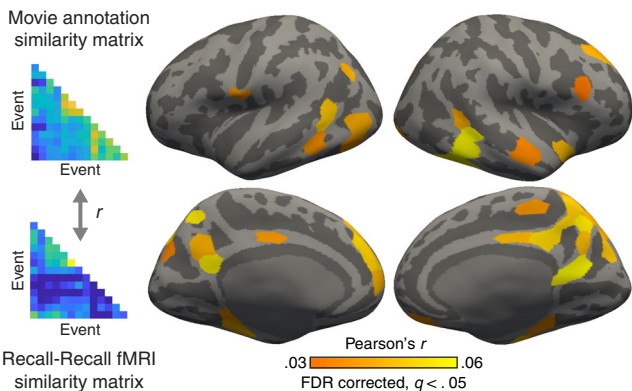

**Fig. 5 Representational similarity during recall.** To identify brain regions whose activation patterns during recall reflect the whole semantic narrative network structure, we performed a representational similarity analysis (RSA[34]). For each cortical parcel, the representational similarity between the fMRI patterns and movie annotations was computed within each movie by correlating the cross-event intersubject pattern similarity matrix and the USE sentence embedding vector similarity matrix. The correlation coefficients were averaged across movies and participants. The resulting mean representational similarity was tested for statistical significance against zero using a randomization test. The whole-brain RSA map was thresholded at $q < .05$ (FDR-corrected across parcels).

PMC and functionally connected areas are engaged in the episodic construction and representation of events or situation models[16].

We next demonstrated that neural patterns in DMN areas reflect not only the situations specific to individual events within each movie, but also the semantic relationships between them during recall. We used whole-brain representational similarity analysis (RSA[34]): for each cortical parcel and movie, we correlated the event-by-event similarity matrix based on the text descriptions of events (i.e., USE vectors from the movie annotations) and the similarity matrix based on neural responses during recall (Fig. 5). The neural similarity was again computed as intersubject pattern correlation, but here the pISC was computed between different events rather than matching events. Statistical significance was determined by randomization tests using event labels randomly shuffled within each movie, and then corrected for multiple comparisons across parcels (FDR $q < .05$). We found positive correlations between the semantic similarity and neural similarity in parcels mostly within DMN, especially those in and around PMC (Fig. 5). We also observed similar but stronger effects in DMN using the semantic similarity matrix generated from participants' recall transcripts rather than movie annotations (Supplementary Fig. 12b).

**Narrative network centrality predicts the between-brain similarity of event representations.** Our next key question was whether the centrality of events modulates the quality of event-specific neural representations in DMN measured as pISC. Here, we used a region-of-interest (ROI) approach (Fig. 4b) and focused on PMC, which showed the strongest effects in the whole-brain pISC and RSA analyses above. As a lower-level control region, we used the early visual cortex (EVC). Both regions showed event-specific neural patterns (i.e., significantly positive pISC) during movie watching (pISC in PMC = .12, EVC = .3, one-tailed randomization $p$s < .001; Fig. 4c, left panel) and recall (pISC in PMC = .06, EVC = .01, $p$s < .001; Fig. 4c, right panel).

For each ROI, we compared the mean pISC of high vs. low semantic centrality events defined within each movie.

Randomization tests were used to test the statistical significance of the difference between conditions. During recall, higher semantic centrality of an event was associated with higher pISC in PMC (high − low difference = .019, two-tailed randomization $p = .037$; Fig. 4e), whereas lower semantic centrality was associated with higher pISC in EVC (difference = −.013, $p = .012$; Fig. 4f). These findings indicate that high semantic centrality events were represented in a more reliable and convergent manner across brains within a higher associative region supporting situation model representations, but not within a sensory control area. In contrast, no significant difference between conditions was observed in either ROI during movie watching, although the direction of effect was consistent with that during recall in both ROIs (PMC difference = .019, $p = .17$; EVC difference = −.031, $p = .12$; Supplementary Fig. 11b). While speculative, the diminished effect of centrality on pISC during movie watching may reflect the fact that the structure of the whole narrative becomes apparent only after participants finish watching the movies (i.e., during recall).

In this and all the above analyses involving pISC during recall, twelve events recalled by fewer than five participants were excluded. However, our main pISC analysis results remained qualitatively identical when all events were included in the analysis (Supplementary Fig. 13).

**Narrative network centrality modulates hippocampal encoding signals.** Hippocampus has been known to play a crucial role in encoding continuous narratives as discrete events[35]. Hippocampus activation increases at the offset of a movie event, and the magnitude of the activation predicts subsequent remembering and neural reactivation of the event[12,15,36]. This boundary response has been interpreted as the registration of the just-concluded event into long-term memory. We tested whether the centrality of events influences the offset-triggered hippocampal encoding signal during movie watching, potentially mediating the behavioral effect of narrative network centrality. We measured the time courses of hippocampal blood oxygenation level dependent (BOLD) responses locked to the boundaries between events, and found that hippocampal responses were higher following the offset of high than low semantic centrality events (Fig. 6a). In contrast, hippocampal responses following the onset of high vs. low centrality events (i.e., before the events fully unfold and diverge in terms of their semantic contents) were not significantly different from each other (Fig. 6b), confirming that semantic centrality specifically affected the encoding of information accumulated during just-concluded events. Stronger hippocampal event offset responses (averaged across 10 – 13 TRs from each offset; TR = 1.5 s) also predicted the successful recall of individual events in a mixed-effects logistic regression analysis ($\beta = .26$, SE = .1, $\chi^2(1) = 6.37$, $p = .012$), consistent with prior studies[12,36]. Moreover, hippocampal offset responses significantly mediated the effects of semantic centrality on event recall (average causal mediation effects = .001, $p = .016$, 95% CI = [.0002, .003]); the effect of semantic centrality was still significant after controlling for hippocampal responses ($\beta = .2$, SE = .05, $\chi^2(1) = 13.91$, $p < .001$), indicating a partial mediation. These results suggest that rich connections between events lead to stronger hippocampus-mediated encoding.

Hippocampus also interacts with higher associative cortices when encoding naturalistic events[37], and increased hippocampus-cortex connectivity during encoding is associated with successful learning and memory formation[38,39]. Does the centrality of events affect hippocampal-cortical coupling as well? We used intersubject functional connectivity analysis (ISFC[32]) to measure the interaction between the hippocampus and cortical ROIs

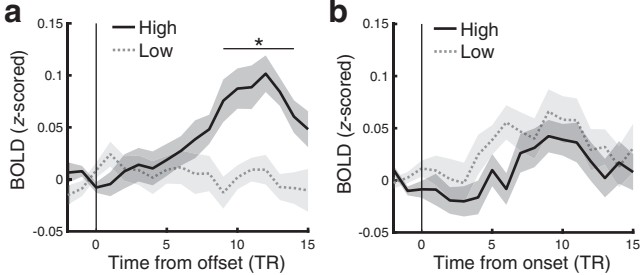

**Fig. 6 Effects of semantic centrality on hippocampal event boundary responses.** Mean hippocampal blood oxygenation level-dependent (BOLD) response time courses aligned at the offset (**a**) or onset (**b**) of events during movie watching. Solid lines and dotted lines show responses for the high and low semantic centrality events, respectively. The BOLD time course of each event was first baseline corrected by subtracting the mean response of the two TRs (TR = 1.5 s) immediately preceding the offset/onset of the event from each time point. The baseline-corrected time courses were averaged across events within each movie and then across movies and participants. Shaded areas indicate SEM across participants. Statistical significance reflects the difference between High vs. Low centrality events at each time point. *$q < .05$ (FDR corrected across time points). Source data are provided as a Source Data file.

during movie watching. ISFC computes correlations between activation time courses of different brain regions across participants rather than within participants, which makes it possible to isolate stimulus-locked activity from background noise[32]. We first computed ISFC between the hippocampus and PMC during the 26 movie events which were 22.5 s (15 TRs) or longer. Functional connectivity patterns computed within windows as short as 22.5 s have previously been shown to robustly predict cognitive states[40]. We then correlated the ISFC values with the semantic centrality of the events. We found that the hippocampal-PMC interaction was stronger for higher centrality events ($r(26) = .49$, $p = .01$, 95% CI = [.13, .74]). In contrast, the hippocampal-EVC interaction did not show a significant relationship with centrality ($r(26) = .01$, $p = .95$, 95% CI = [−.38, .4]), and the correlation was significantly lower than that between hippocampus-PMC ISFC and centrality (95% CI of the difference between correlations[41] = [.05, .87]). Similar results were observed using different minimum event duration thresholds (Supplementary Table 5). The stronger hippocampal-PMC connectivity during higher centrality events might reflect greater reinstatement of other event representations cued by overlapping components (e.g., ref. [42]). However, due to the limited number of movie events included in the analysis, it will be important to replicate these findings with a larger dataset.

## Discussion

In this study, we found that the structure of inter-event connections in complex naturalistic experiences predicts the behavioral and neural signatures of their memory traces. We applied an approach of transforming audiovisual movies into networks, whose nodes are events and whose edges are based on semantic similarity or cause-effect relationships between events. Participants watched and recounted the movies in their own words; events highly connected with other events within the narrative network, i.e., high centrality events, were more likely to be recalled. Higher centrality was also associated with greater hippocampal activity at event boundaries, as well as with increased hippocampal-cortical interaction during movie watching. Furthermore, recalling high centrality events more strongly recruited high-order cortical regions in the DMN involved in episodic recollection, and the multivoxel patterns of high-centrality events

were reinstated in a more convergent manner across individuals, relative to lower-centrality events. These findings demonstrate that the specific structure of relations between events in a natural experience predicts both what will be remembered and what the properties of hippocampal and DMN regions will be during later recollection.

Recent years have seen an explosion in the use of naturalistic stimuli such as movies and narratives in exploring the behavior and neuroscience of human memory, as they provide an engaging laboratory experience with strong ecological validity compared to isolated words or pictures[7,8,43]. These studies have suggested that findings from traditional random-item list paradigms, which have dominated the field for decades, do not always fully extend to naturalistic recall (e.g., ref. [26]). In line with this, we observed that the recall probability of events from a movie does not show serial position effects typically reported in random-item list learning[3,28] where the first and last few items in a list tend to be better remembered than items in the middle. This finding was consistent regardless of whether each participant watched a single movie (Supplementary Fig. 7c) or a series of movies in a row (Fig. 2d). The lack of clear primacy or recency advantages may be due to the inter-event dependencies which made each narrative a coherent structure, supporting memories for central events which did not necessarily occur at the beginning or end of the story; that is, inter-event connections may overshadow the existing effects of temporal positions. At the same time, centrality effects may not be specific to narratives; semantically related items in a random list trigger recall of each other[4,44], which could lead to better memory performance for those items. Furthermore, the event complexity of narratives is not likely to be the main reason for the lack of serial position effects: in a highly realistic encoding setting (a real-world walk) where the events consisted of unrelated activities (e.g., visiting a number of artworks), naturalistic recall showed similar characteristics to random-item list recall[45]. Further studies are needed to determine the roles of structural coherence and event complexity in centrality effects on memory. Understanding when and how classic list-based memory effects fail to extend to narratives and other natural stimuli will be essential for evaluating the results of future studies which use complex realistic conditions to study memory.

Why does higher event centrality improve memory performance? Distinct benefits may be present at recall and at encoding. During recall, high centrality events may have greater opportunity to be cued during recall, as by definition they have higher association strengths with other events. In our experiment, recall itself took the form of a narrative; under these conditions, high centrality events may have been especially likely to be recounted, because omitting them might disproportionately affect the logical structure and coherence of the reconstructed story. Inter-event connections may also benefit encoding. During movie watching, events highly connected with other events are more likely to reactivate and be reactivated by the other events containing shared components[42,46,47]. Consistent with this, we found that the coupling between the hippocampus and a cortical region involved in representing events (PMC[16,48], see below) was stronger when participants were watching events with higher semantic centrality. The reactivation of high centrality events during encoding may result in more robust memory for those events by functioning as repeated encoding, and/or integrating the interconnected events to form joint representations[19,20,49]. The benefit of high centrality during encoding is also reflected in the greater hippocampal responses following the offset of high than low centrality movie events (Fig. 6a). Such hippocampal event boundary responses have been linked to the successful registration of just-concluded episodes into long-term memory[12,13,35], which was replicated in the current study. It

has been shown that DMN connectivity during movie-viewing is modulated by surprise[50]; one possibility is that the conclusion of a higher centrality event produces greater uncertainty in the ongoing narrative, as higher centrality events are more likely to influence the main storyline of the narrative. This may result in a more salient boundary and stronger boundary-evoked encoding signals.

We demonstrated that DMN activity during remembering was modulated by the recollected event's position in the narrative network. High-level associative areas in the DMN[14], especially the PMC and its functionally connected subregions such as the angular gyrus, have been implicated in the episodic construction and representation of events[16,17,48]. In accordance with this view, we observed event-specific neural activation patterns in the medial and lateral DMN areas during recall (Fig. 4d), and representational similarity analysis revealed that the relational structure of these neural event patterns could be predicted by human-generated descriptions of the movie and by recall transcripts (Fig. 5, Supplementary Fig. 12b; for a similar approach, see ref. [26]). Critically, activation in the PMC and angular gyrus scaled with the degree to which events had more connections with other events during recall (Supplementary Fig. 9b, 9d), consistent with prior studies showing that these areas are involved in combining and comprehending semantically connected information[51–53].

Furthermore, higher semantic centrality predicted greater between-participant pattern convergence in PMC (Fig. 4e). This is likely to be a neural signature of stronger and more accurate recall of episodic details[17,54] for high centrality events, dovetailing with the behavioral results. Additionally, higher intersubject similarity for high centrality events might arise from design pressure on narratives. Highly connected events are likely to be logically important in a story; indeed, we found that semantic centrality was positively correlated with the perceived importance of events as retrospectively rated by independent coders ($r(202) = .22$, $p = .002$, 95% CI = [.08, .34]). Thus, to aid the understanding of their linked events and eventually the whole story, high centrality events need to be designed in a way that minimizes the variability or ambiguity in how people interpret them. This adoption of a similar canonical interpretation of an event across people gives rise to more similar neural responses across individuals[55–57]. The design pressure may even produce unique characteristics associated with high centrality shared across events and narratives; high centrality events were semantically more similar to other high centrality events than to low centrality events across movies, although the difference was small (difference in $r = .047$, $p < .001$). Future work may investigate whether real-life everyday events without such design pressure would show similar centrality effects to what we observed here using fictional narratives.

In contrast with the pattern in the DMN, we observed that a low-level sensory region (EVC) showed higher between-brain convergence for low semantic centrality events during recall (Fig. 4f). This result should be interpreted with caution as the overall pISC was extremely low in EVC during recall due to the absence of shared visual stimulation (below the level typically considered reliable signal, in line with prior reports[17,18,58]). Nonetheless, we can speculate that the opposite effects obtained in PMC and EVC may reflect switching between internal and external modes of processing, primarily involving higher-order cortices in the DMN and sensory areas, respectively[59,60]. Participants are more likely to be in internal mode that prioritizes retrieval[61,62] while watching high centrality events that reactivate associated events, whereas external mode is more likely to prevail during low centrality events as participants would focus primarily on the novel current input. This may result in more visually-driven memory reinstatement (e.g., involving salient visual fragments rather than the gist of the event) and thus stronger pISC in

EVC for low centrality events. EVC indeed tended to show higher activation for low semantic centrality events during movie watching (Supplementary Fig. 9a), even though low-level visual features such as luminance and contrast were not modulated by semantic centrality. Similarly, a recent study[63] reported that the visual sensory network is more activated when participants report a lower understanding of an ongoing narrative.

One might have expected that the effects of narrative structure would not be apparent in brain responses measured during ongoing movie watching, as the full structure of inter-event connections is only available after all movie events are completed. Still, as discussed above, event centrality significantly influenced hippocampal and cortical univariate responses during movie watching. A simple explanation for these results is that centrality based on partial narrative networks (i.e., a network that excluded events not-yet-presented) was sufficiently similar to the full-narrative centrality values, especially later in a movie. Indeed, semantic centrality computed from networks excluding not-yet-presented events was positively correlated with that based on full networks ($r(192) = .76$, $p < .001$, 95% CI = [.69, .81]). Another interesting, and not mutually exclusive, possibility is that participants were able to predict the full-narrative centrality of a current event by anticipating the potential connections with future events. In support of this interpretation, we found a strong positive correlation between the perceived importance of events obtained concurrently with watching the movies and those obtained retrospectively, after the movie ended ($r(202) = .67$, $p < .001$, 95% CI = [.58, .74]). Predictions of event centrality could be based on the learned schema of canonical story structures[64,65] as well as on director's cues used in popular movies such as luminance and shot motion[66]. Future work will explore how brain responses are driven by the temporally evolving, rather than static, inter-event structure when participants consume unpredictable stories, or actively engage in selecting upcoming narrative events. Future work may also explore the cognitive and neural mechanisms supporting the learning of novel narrative network structures, and whether they are similar to learning the network structure of simple isolated stimuli or actions (e.g., refs. [67,68]).

What is the nature of information reflected in narrative network centrality? We believe that both semantic centrality and causal centrality primarily reflect aspects of how situation models for individual events are related to each other, rather than surface features (or "textbase") of the stimuli or annotations as discussed in classic discourse theories[69,70]. With respect to semantic centrality, each event annotation included descriptions from three different annotators and often consisted of multiple sentences. All of these different sentences and phrasing choices were incorporated into the text embedding for each event, from which the semantic narrative network arises. Thus, the event embedding vectors capture information abstracted beyond the surface features of the original sentences. With respect to causal centrality, cause-effect relations between events are traditionally associated with the situation model, identified as crucial elements of knowledge structures[69,71,72]. Indeed, half of the movies in the current study contain no dialogue at all, and thus human raters cannot be relying on text or language provided in the stimulus to make causality judgments. Thus, while some surface features such as annotators' word choices should be expected to reflect aspects of the situation model (see Supplementary Methods for semantic centrality based on word-level information), situation-level information, rather than low-level textual overlap, is likely to determine our centrality measures.

Causal relations have long been considered an important organizing factor for event and narrative memories[22,64,73]. Consistent with earlier work, we found that events with stronger

causal connections with other events are better remembered (Fig. 3c, d), and these effects were not redundant to those of semantic connections. Yet, while the effects of causality on univariate responses during movie-viewing (see also ref. [63]) were comparable to the effects of semantic centrality (Supplementary Figs. 9c and 10c), multivoxel pattern effects of causality during recall were not as clear as those of semantic similarity (Supplementary Figs. 10b, e). Several characteristics of causal relations in movie stimuli might have reduced the reliability of the effects of causal narrative network structure. First, causal relations were sparse and mostly identified between adjacent events (Supplementary Figs. 4a, 5d). In addition, causality judgments may be more idiosyncratic: average across-coder correlation was lower for causal (mean $r(202) = .34$) than semantic centrality (mean $r(202) = .52$) when centrality was computed from each individual coder's causality rating or movie annotation. It is also noteworthy that semantic and causal connections were measured in distinct ways (text embeddings and human judgments, respectively) and reflect different types of information: semantic connections are based on similar or shared features such as people, places, and objects, whereas causal connections additionally require an action, its outcome, and internal models providing a logical dependency between the two[74,75]. For example, two events "Jill threw the ball" and "Jack fell to the ground, unconscious" may have a clear causal link for a reader with some background knowledge, but low semantic similarity according to text embeddings as they have no overlapping or similar-meaning words or topics. In this study, we did not focus strongly on dissociating semantic and causal centrality, as they were positively correlated in our movie stimuli. Future studies designed to orthogonalize different types of inter-event relations, including semantic and causal relations as well as other dimensions such as emotional similarity[76,77], will be able to further clarify their unique influences on the behavioral and neural signatures of memory. Additionally, further investigations with more stimuli may examine the extent to which the two co-occur in naturalistic narratives, as well as in non-narrative real-world experiences.

In summary, we applied a recently developed natural language model and neuroimaging techniques to a universal and natural form of human memory recall: telling stories about the past. This approach allowed us to demonstrate that rich connections between events in complex realistic experiences protect against forgetting and predict the neural responses associated with successful memory encoding, as well as the properties of brain activity during spoken recall. Consideration of the effects of inter-event structure on real-world memory may benefit practical applications such as the development of memory interventions for clinical and healthy aging populations[78] or promoting learning in educational settings[79–81]. In addition, our work demonstrates that holistic metrics which capture the interrelations of events within episodes may be important to incorporate into models of learning and comprehension, especially as these models grow in their sophistication and power to explain complex experiences in the real world[82,83].

## Methods

The current study complies with ethical regulations for research on human participants. The fMRI experiment was conducted following protocols as approved by the Princeton University Institutional Review Board. The preregistered online experiment was conducted following protocols as approved by the Johns Hopkins University Institutional Review Board.

**Participants**. Twenty-one healthy participants were recruited from the Princeton community (12 females, age 20–33 years, mean age 26.6 years). All participants were right-handed native English speakers and reported normal hearing and normal or corrected-to-normal vision. Informed consent was obtained in accordance with procedures approved by the Princeton University Institutional Review

Board. Participants received monetary compensation for their time ($20 per hour). Data from 6 of the 21 participants were excluded from analyses due to excessive head motion (absolute displacement greater than 4 mm) in at least one scanning run.

**Stimuli**. The audiovisual stimuli consisted of 10 short movies including 3 animations and 7 live-action movies. The movies were on average 4.54 min long (ranged 2.15 to 7.75 min) and had narratives that varied in content and structure. Two of the movies consisted of short clips edited from longer full movies (Catch Me If You Can, The Prisoner). Detailed information about each movie is provided in Supplementary Table 1. Each movie was prepended with a 6-s long title scene in which the title in white letters appeared at the center of the black screen and then gradually disappeared. Five movies were presented in the first movie watching phase scanning run and the other five were presented in the second run. The movies were played consecutively within each scanning run without gaps in between other than the title scenes. The presentation order of the ten movies was fixed across participants. As in our prior study[17], an additional 39-s audiovisual cartoon (Let's All Go to the Lobby) unrelated to the movie stimuli was prepended at the beginning of each movie watching scanning run. The introductory cartoon was excluded from analyses.

**Experimental procedures**. The experiment consisted of three phases: movie watching, free spoken recall, and cued spoken recall. All three phases were performed inside the scanner. Before the movie watching phase, participants were told that they would be watching a series of short movies. As in our prior study[17], we instructed participants to pay attention to the movies as they would normally do in real life. Participants were also told that they would be asked to verbally describe the movie plots later. The movie watching phase consisted of two consecutive scanning runs. Participants watched five movies in each run (first run video duration = 24.9 min, second run video duration = 22.9 min). No behavioral responses were required from the participants during scanning.

The free spoken recall phase immediately followed the movie watching phase. Participants were instructed to describe aloud what they remembered from the movies in as much detail as they could, regardless of the order of presentation. We encouraged participants to speak for at least 10 min and told them that if they chose to speak for longer, that would be even better. Participants were also allowed to return to a movie that they had described earlier in case they realized they had missed something while speaking about another movie. We instructed participants to verbally indicate that they were finished by saying "I'm done" after recalling everything they could remember. A white fixation dot was presented on the black screen while participants were speaking; participants were told that they did not need to fixate on this dot. In case participants needed to take a break or the duration of the scanning run exceeded the scanner limit (35 min), we stopped the scan in the middle and started a new scanning run where participants resumed from where they had stopped in the previous run. 4 of the 15 participants included in the analysis had such a break within their spoken recall session.

During the cued spoken recall phase immediately following the free spoken recall phase, participants viewed a series of titles of the ten movies they watched. For each movie, participants were instructed to first read the title out loud and then describe the movie. Participants were told to provide a short summary of a few sentences in case they previously described the movie during the free spoken recall, but describe the movie in as much detail as they could if the movie was previously forgotten. The cued spoken recall phase was not analyzed for the current study.

All visual stimuli were projected using an LCD projector onto a rear-projection screen located in the magnet bore and viewed with an angled mirror. The Psychophysics Toolbox (http://psychtoolbox.org/) for MATLAB was used to display the stimuli and to synchronize stimulus onset with MRI data acquisition. Audio was delivered via in-ear headphones. Participants' speech was recorded using a customized MR-compatible recording system (FOMRI II; Optoacoustics Ltd.).

### Behavioral data collection and preparation

*Movie event segmentation*. Each of the ten movie stimuli was segmented into 10–35 events (mean 20.2, excluding the title scenes) by an independent coder who was not aware of the experimental design or results. Following the method used in our previous study[17], we instructed the coder to identify event boundaries based on major shifts in the narrative (e.g., location, topic, and/or time). Unlike in the prior study that used a 50-min movie[17], we did not set the minimum event duration (10 s) because we used much shorter movie stimuli in the current study. The coder gave each event a descriptive label (for example, "girl inside room alone with a pizza"). The start and stop timestamps of each event were recorded. There were 202 movie events in total and the duration of events ranged from 2 to 42 s (s.d. = 7.4 s). The number and the mean duration of events for individual movies are summarized in Supplementary Table 2.

*Movie annotations*. Movie annotations were provided by three independent annotators who did not participate in the fMRI experiment. Each annotator identified finer-grained sub-event boundaries within each of the 202 movie events based on their subjective judgments. The beginning and end of the fine-grained

sub-events were also timestamped. For each sub-event, the annotators provided written descriptions about what was happening in the movie at that moment in their own words. No edits were made on the written descriptions other than correcting typos and removing/replacing special characters not recognized by our text analysis scripts. Supplementary Table 2 summarizes the number of fine-grained sub-events and the number of words generated by individual annotators for each movie. An example movie annotation can be found in Supplementary Table 3.

*Recall transcripts*. The audio recording of each participant's free spoken recall was transcribed manually. Each recall transcript was segmented into discrete utterances based on pauses and changes in the topic. The recall transcripts were segmented such that each utterance was not longer than 50 words. Timestamps were also identified for the beginning and end of each utterance. Each utterance was categorized as one of the followings based on its content: (1) recall of specific movie events, (2) general comment about the movie, (3) memory search attempt (e.g., "Let's see…"), (4) end of recall (e.g, "I'm done."), and (5) speech unrelated to the task (e.g., "Can I start now?"). In case an utterance was a recall of movie events, the specific movie events described in the utterance were identified. Among the different types of utterances, only the recall of specific movie events was used in the behavioral and fMRI analyses in the current study.

*Importance ratings*. Importance ratings for each of the 202 movie events were collected from four independent raters who did not participate in the fMRI experiment. The raters watched each movie and then retrospectively rated how important each event was for understanding what happened within the movie on a scale from 1 (not important at all) – 10 (very important). The ratings were averaged within each event across raters for analyses (range 1.5–10 across events, mean 6.09, s.d. 1.92). We additionally collected importance ratings from a separate group of four independent raters while they were watching the 10 movies for the first time. At the end of each movie event, the movie stopped playing, and the raters rated the importance of the just-played event on a scale from 1 to 10. These rate-as-you-go importance ratings averaged across the raters were positively correlated with the retrospective ratings ($r(202) = .67$, $p < .001$, 95% CI = [.58, .74]). Importance ratings were positively correlated across raters for both retrospective ratings and rate-as-you-go ratings (mean event-wise cross-rater correlation computed within each movie = .65 and .55, respectively).

**Narrative networks**. To quantify and assess the inter-event structure of the movie stimuli, we transformed each movie plot into a graph/network. In this narrative network, the events within a movie (nodes) form connections with each other (edges), and the connection strength between a pair of events (edge weight) is determined by their content similarity or causality. The narrative network edges were unthresholded (except for the visualization of semantic narrative networks) and undirected. The centrality of each individual event within a movie was defined as the degree of each node (i.e., the sum of the weights of all edges connected to the node) in the network, normalized by the sum of degrees and then *z*-scored within each movie. Events with stronger and greater numbers of connections with other events had higher centrality.

*Semantic narrative networks*. Movie annotations were used to generate narrative networks based on the semantic similarity between events (Fig. 1). For each annotator and movie, the text descriptions for the fine-grained sub-events were concatenated within each movie event. The text descriptions were then encoded into high-dimensional vectors with Google's Universal Sentence Encoder (USE[25]), a natural language processing model built in TensorFlow (https://www.tensorflow.org), such that each movie event was represented as a 512-dimensional vector. The USE vectors from the three annotators were highly similar to each other (mean event-wise cross-annotator cosine similarity between all possible annotator pairs = .78; Supplementary Fig. 1); thus the USE vectors were averaged across annotators within each movie event. For each movie, the narrative network was generated by using the cosine similarity between the USE vectors of movie event pairs as the edge weights between nodes (events). The semantic centrality values based on USE sentence embedding vectors were correlated with those based on word-level overlap or word2vec embeddings (Supplementary Methods).

*Causal narrative networks*. To generate narrative networks based on the causal relationship between events (Supplementary Fig. 4), we had 18 independent coders identify causally related event pairs (the cause event and the effect event) within each movie. Each coder coded different subsets of the ten movies and each movie was coded by 12 (Catch Me If You Can) or 13 coders (all the other movies). The coders watched the movies and were given the movie annotation with sub-event segmentation by the annotator JL. The coders were instructed to consider two movie events as causally related if any fine-grained sub-event of an event is a strong cause of any (at least one) sub-event of the other event (see Supplementary Methods for the exact instructions given to the coders). Whether a causal relationship was strong enough to be identified depended on the coders' subjective criteria; the coders were instructed to keep the criteria as consistent as possible. The coders were also told to ignore any causal relationship between the sub-events

within the same event. Thus, an event pair always consisted of two different events. For each movie, the edge weights between nodes in the narrative network were defined as the proportion of coders who identified a movie event pair as causally related, regardless of the cause-effect direction. However, causal centrality computed from directed networks which accounted for the cause-effect direction showed highly similar behavioral effects as the centrality computed from undirected networks (Supplementary Fig. 6). The average Jaccard similarity between a pair of coders' lists of causally related event pairs was .31 (computed within each movie and then averaged across movies).

**Semantic similarity between events across movies**. To examine whether there are semantic characteristics shared among high semantic centrality events across different movies, we computed similarity between event-specific USE vectors (averaged across annotators) across movies. Specifically, we tested whether the similarity between high centrality events was higher than the similarity between high and low centrality events. High and low centrality events were defined as the events whose semantic centrality values were within the top and bottom 40% in each movie, respectively. For each movie, we computed Pearson correlations between the USE vector of each high centrality event and the USE vectors of each of the other movies' high centrality events. The correlation coefficients were averaged across events and movies to produce the mean similarity value for high centrality-high centrality event pairs. Likewise, we computed the mean similarity between each movie's high centrality events and each of the other movies' low centrality events. We then performed a randomization test to assess whether the difference between the mean similarities of high-high pairs and high-low pairs was significantly different from zero. A null distribution of the difference of mean USE vector similarities was generated by randomly shuffling the high or low centrality labels of the events within each movie and then computing the difference 1000 times. A two-tailed *p*-value was defined as the proportion of values from the null distribution equal to or more extreme than the actual difference.

**Mixed-effects logistic regression analysis of recall behavior**. We performed a mixed-effects logistic regression analysis implemented in R's "lme4" package to test the unique effect of semantic centrality and causal centrality on recall performance after controlling for each other. Each event from each participant served as a data point. Data were concatenated across all participants. The dependent variable of each data point was the event recall success (1 = recalled, 0 = not recalled). Normalized semantic and causal centrality were included as fixed effects. Individual participants and movie stimuli were included as random effects. Statistical significance of the unique effect of each type of centrality was tested by performing a likelihood ratio test for the full model against a null model including all independent variables except for the variable of interest.

**fMRI acquisition**. fMRI scanning was conducted at Princeton Neuroscience Institute on a 3 T Siemens Prisma scanner with a 64-channel head/neck coil. Functional images were acquired using a T2*-weighted multiband accelerated echo-planar imaging (EPI) sequence (TR = 1.5 s; TE = 39 ms; flip angle = 50°; acceleration factor = 4; shift = 3; 60 oblique axial slices; grid size 96 × 96; voxel size $2 \times 2 \times 2$ mm$^3$). Fieldmap images were also acquired to correct for B0 magnetic field inhomogeneity (60 oblique axial slices; grid size 64 × 64; voxel size $3 \times 3 \times 2$ mm$^3$). Whole-brain high-resolution anatomical images were acquired using a T1-weighted MPRAGE pulse sequence. Scanning parameters for the anatomical images varied across participants (15 participants had 176 sagittal slices with voxel size $1 \times 1 \times 1$ mm$^3$; 6 participants had 192 sagittal slices with voxel size $.9 \times .86 \times .86$ mm$^3$), as the anatomical images of a subset of participants were originally obtained for other projects unrelated to the current study.

**fMRI preprocessing**. Preprocessing of high-resolution anatomical images and cortical surface reconstruction were performed using FreeSurfer's recon-all pipeline. For each scanning run, functional images were corrected for head motion and B0 magnetic inhomogeneity using FSL's MCFLIRT and FUGUE, respectively. Functional images were then coregistered to the anatomical image, resampled to the fsaverage6 template surface (for cortical analysis) and the MNI 305 volume space (for subcortical analysis), and then smoothed (FWHM 4 mm) using the FreeSurfer Functional Analysis Stream. The smoothed functional data were then high-pass filtered within each scanning run (cutoff = 140 s). For intersubject functional connectivity analysis, we additionally projected out the following nuisance regressors from the filtered functional data: the average time courses (*z*-scored within each run) of 1) high s.d. voxels outside the grey matter mask (voxels in the top 1% largest s.d.), 2) cerebrospinal fluid, and 3) white matter[32]. The resulting time series were *z*-scored within each vertex or voxel across TRs. The first 2 TRs of movie watching scanning runs were discarded as the movies were played 2 TRs after the scanning onset. The first 3 TRs of both movie watching and free spoken recall scanning runs were additionally removed, shifting the time-courses by 4.5 s, to account for the hemodynamic response delay.

**Cortical parcellation and region of interest (ROI) definition**. For whole-brain pattern-based analyses, we used a cortical parcellation atlas based on fMRI functional connectivity patterns[33]. Specifically, we used the atlas where the cortical

surface of the brain is divided into 400 parcels (200 parcels per hemisphere) which are clustered into previously reported 17 functional networks[84]. For region-of-interest analyses, we defined the bilateral posterior-medial cortex (PMC) and the bilateral early visual cortex (Fig. 4b) by combining parcels from the 400-parcel atlas that correspond to the areas of interest. The PMC ROI consisted of the posterior cingulate cortex and precuneus parcels in the default mode network. The early visual cortex ROI consisted of parcels around the primary visual cortex (see Supplementary Table 4 for the list of parcels used to create the ROIs). The bilateral hippocampus mask was extracted from FreeSurfer's subcortical (Aseg) atlas on the MNI volume space.

**Univariate activation analysis**. We performed whole-brain univariate activation analysis on the cortical surface to identify regions whose activation scales with the narrative network centrality (Supplementary Fig. 9). The analysis was performed separately for the movie watching phase and the recall phase. For each vertex of each participant, we first computed the mean activation for each movie event by averaging the preprocessed BOLD signal across TRs that correspond to the event. The first event of each movie was excluded from this and all other univariate analysis of the movie watching phase (see Supplementary Fig. 8). For the recall phase, only the events successfully recalled by the participant were included in the analysis. We then performed a linear regression where the event-by-event activation (combined across all 10 movies) was explained by the semantic or causal centrality of the events, after regressing out the overall movie-level activation from the event-by-event activation. Finally, one-sample $t$-tests against zero (two-tailed) were applied on the participant-specific vertex-wise parameter estimate maps to generate the group-level $t$-statistic map.

We also compared the ROI-specific univariate activation for high vs. low centrality movie events during each experimental phase. High and low centrality events were defined as the events whose semantic/causal centrality metrics were within the top and bottom 40% in each movie, respectively. For each participant and event, the preprocessed BOLD signals were first averaged across voxels or vertices within an ROI and across all TRs corresponding to the event. The mean signal was then averaged across events in the same condition and then across movies, resulting in a single value per participant and condition. Two-tailed paired $t$-tests were used to test the statistical significance of the difference between the high vs. low centrality conditions.

**Intersubject pattern correlation analysis**. Whole-brain intersubject pattern correlation (pISC[17,31]) maps were generated for the movie phase (Movie-Movie similarity; Supplementary Fig. 11a) and the recall phase (Recall-Recall similarity; Fig. 4d) separately. pISC was calculated in a participant-pairwise manner using the following procedures. For each cortical parcel of each participant, first the mean activation pattern of each event was generated by averaging the preprocessed movie or recall phase BOLD data across TRs within the event in each vertex within the parcel. Note that as recall BOLD data existed only for successfully recalled events, each participant had a different subset of recall event patterns. For each participant and event, we computed the Pearson correlation between the event pattern of the participant and the pattern of the matching event from each of the remaining participants, which resulted in $N - 1$ correlation coefficients ($N$ = the total number of participants who watched/recalled the event). The correlation coefficients were then averaged to create a single pISC ($r$) value per event per participant. These pISC values were averaged across events (combined across all 10 movies) and participants, resulting in a single pISC value for each parcel. We performed a randomization test for each parcel to test the statistical significance of the mean pISC. Parcel-wise mean pISC values were obtained using the same procedures as described above, except that we randomly shuffled the event labels before computing the between-participants pattern similarity. That is, one participant's neural pattern of an event was correlated with another participant's neural pattern of a non-matching event. This procedure was repeated 1000 times to generate a null distribution of pISC. A one-tailed $p$-value was defined as the proportion of values from the null distribution equal to or greater than the actual mean pISC. The $p$-values from the entire cortical surface were corrected for multiple comparisons across all 400 parcels using the Benjamini–Hochberg procedure ($q < .05$).

We also computed pISC in the PMC and early visual cortex to test the relationship between the semantic/causal narrative network centrality metrics and event-specific neural representations in the ROIs. The participant-specific, event-by-event pISC values were computed for each ROI in the same way we computed pISC for each parcel of the whole-brain pISC map above (Fig. 4a), separately for movie watching and recall. We compared the pISC for high vs. low centrality events, defined as the events whose centrality metrics were within the top or bottom 40% in each movie. The pISC values were first averaged across events within the high/low centrality condition for each movie and then across movies, resulting in a single pISC value per condition per participant. We then ran a randomization test to assess whether the difference of pISC between the high vs. low centrality conditions, averaged across participants, was significantly different from zero. A null distribution of the mean difference between the conditions was generated by randomly shuffling the event labels of the event-specific pISC values within each movie and then computing the difference 1000 times. A two-tailed $p$-value was defined as the proportion of values from the null distribution equal to or more extreme than the actual difference.

In all analyses involving intersubject neural similarity (including the representational similarity analysis and the intersubject functional connectivity analysis), the first event of each movie was excluded from movie phase analyses to minimize the effect of movie onset (Supplementary Fig. 8). For recall phase analyses, we excluded twelve events recalled by fewer than five participants (1–3 events per movie from 6 movies). However, we obtained qualitatively identical results when we included all events in the analyses (Supplementary Fig. 13).

**Representational similarity analysis**. We performed representational similarity analysis[34] by comparing the event-by-event similarity matrices based on two different types of event representations: the text descriptions of events (i.e., movie annotations or recall transcripts) and neural activation patterns measured during recall (Fig. 5, Supplementary Fig. 12b). The similarity matrix based on movie annotations was generated for each movie by computing the pairwise cosine similarity between the USE vectors of all events within the movie. This matrix was identical to the adjacency matrix of the semantic narrative network. To create the similarity matrix based on recall, we first extracted the sentences from each participant's recall transcript describing each event and then converted them into USE vectors. The similarity matrix was generated for each participant and movie by computing the cosine similarity between the USE vectors of all events recalled by the participant. The matrices were then averaged across all participants. As participants recalled different subsets of events, the number of participants averaged was different across event pairs.

The fMRI recall pattern similarity matrix was generated for each parcel of the Schaefer atlas. Within each of the ten movies, we computed pattern correlations (Pearson $r$) between all possible pairs of events between all pairs of participants. For each participant and movie, this resulted in $N - 1$ fMRI pattern similarity matrices with the size of $M \times M$, where $N$ is the total number of participants and $M$ is the number of events within the movie. We took the average of each matrix and its transpose to make the similarity matrix symmetric (i.e., similarity between events $a$ and $b$ across participants $i$ and $j$ = average of corr(participant $i$ event $a$, participant $j$ event $b$) and corr(participant $j$ event $a$, participant $i$ event $b$)), and then averaged the $N - 1$ similarity matrices to generate a single fMRI similarity matrix per movie and participant.

The representational similarity between a text-based similarity matrix and an fMRI pattern-based similarity matrix was measured by computing the Pearson correlation between the lower triangles (excluding the diagonal values) of each matrix. The correlation coefficients were next averaged across movies and then across participants to create a single value per parcel. For each parcel, a randomization test was performed to test whether the mean representational similarity was significantly greater than zero. We randomly shuffled the event labels of the text-based similarity matrix within each movie and then computed the mean representational similarity as described above. This procedure was repeated 1000 times to generate a null distribution, and a one-tailed $p$-value was defined as the proportion of values from the null distribution equal to or greater than the actual mean representational similarity. The whole-brain $p$-values were corrected for multiple comparisons across parcels using the Benjamini–Hochberg procedure ($q < .05$).

**Hippocampal event boundary responses**. We compared hippocampal event boundary responses following the onset/offset of high vs. low centrality events during movie watching (Fig. 6), High and low centrality events were defined as the events whose centrality values were within the top or bottom 40% in each movie. We first averaged TR-by-TR BOLD signals across voxels within the bilateral hippocampus mask for each participant. We then extracted time series around the onset/offset ($-2 - 15$ TRs) of each high/low centrality event. The first and last events of each movie were excluded to minimize the effect of between-movie transitions. Each time series was baseline corrected by subtracting the mean activation of the two TRs immediately preceding the onset/offset of the event from each time point. The participant-specific time series were then averaged across events within each condition and then across movies. Two-tailed paired $t$-tests were used for each time point to compare the high vs. low centrality conditions. We applied the Benjamini-Hochberg procedure ($q < .05$) to correct for multiple comparisons across time points.

To test whether the effect of semantic centrality on event-by-event recall success (1 = recalled, 0 = not recalled) was mediated via the hippocampal event offset responses, we performed a mediation analysis. Each event from each participant served as a data point, and data were concatenated across all participants. For each participant, the hippocampal offset response of each event was computed by averaging the BOLD time series measured from 10 to 13 TRs after the event offset. Again, the responses were baseline corrected for each event by subtracting the mean response of the two TRs immediately preceding the event offset from the time series. The first/last events of each movie and not recalled events were excluded from the analysis. Three mixed-effects linear or logistic regression models were defined to test 1) the total effect of semantic centrality on recall success (logistic), 2) the effect of semantic centrality on hippocampal offset responses (linear), and 3) the direct effect of semantic centrality on recall success, controlling for hippocampal offset responses (logistic). An additional mixed-effects logistic regression analysis was also performed to test the effect of hippocampal offset responses on recall success. In all models, participants were included as random

effects. The significance of the indirect effect of hippocampal offset responses on the relationship between semantic centrality and recall success was tested via the quasi-Bayesian Monte Carlo simulation as implemented in the "mediation" package in R. Specifically, 1000 simulations were performed to compute the 95% confidence interval of the average causal mediation effects.

**Intersubject functional connectivity analysis**. We performed intersubject functional connectivity analysis (ISFC[32]) to test the relationship between narrative network centrality and the hippocampus-cortex interaction during movie watching. We first averaged the TR-by-TR time courses of the preprocessed (non-neuronal signals removed; see fMRI preprocessing) functional data across all voxels/vertices within each of the hippocampal and cortical ROIs (PMC, early visual cortex). For each movie event as long as 22.5 s or longer (total number of events used across all movies = 26), we computed the ISFC between the hippocampus and a cortical ROI. Functional connectivity patterns computed within windows as short as 22.5 s have previously been shown to robustly predict cognitive states[40]. For each participant, we correlated the participant's hippocampal time series of the event and the cortical ROI time series averaged across all other participants. We then averaged the Pearson correlation coefficients across all participants. This procedure was repeated by correlating each participant's cortical ROI time series and the hippocampal time series averaged across all other participants. Again, the correlation coefficients were averaged across participants. We then took the mean of the two averaged correlations to produce a single ISFC between the hippocampus and the cortical ROI for each event. Finally, we computed the Pearson correlation between the event-wise ISFC and the semantic/causal narrative network centrality.

**Low-level sensory characteristics of movie events**. We measured the low-level visual and auditory features of movie events to examine whether the sensory characteristics can explain the effects of centrality on neural responses during movie watching. For visual features, we measured luminance and contrast averaged across grayscale-converted movie frames within each event. In each frame, luminance was defined as the mean of pixel values, and contrast was defined as the difference between the maximum and minimum pixel values. For an auditory feature, we measured the mean amplitude of sounds played during each event. We extracted the single-channel downsampled (8000 Hz) version of audio signals from the movie clips. Within each event, the audio signals were divided into 100-ms segments, and each segment's amplitude was computed as the difference between the maximum and minimum signal intensities. The amplitudes were then averaged across all segments within each event. The first events of each movie were excluded from the analysis to be consistent with the movie phase fMRI analyses. All sensory features were z-scored across events. We performed mixed-effects linear regression analyses to test whether each of the event-wise low-level sensory features was modulated by semantic centrality, using semantic centrality as a fixed effect and movies as a random effect. Statistical significance of the effect of centrality was tested via likelihood ratio tests for the full models against the null models including the random effect of movies only.

**Preregistered online experiment**. We conducted an online experiment to replicate and generalize the behavioral results of the fMRI experiment with a larger number of participants and a new set of movie stimuli (Supplementary Fig. 7). The online experiment was preregistered at AsPredicted (https://aspredicted.org/fw59g.pdf; Preregistration date: July 15, 2019). We recruited a total of 393 participants (194 female, 198 male, 1 other) on Amazon's Mechanical Turk using the psiTurk system[85]. The final sample size was slightly larger than the preregistered sample size (N = 380) because one extra batch of participants was recruited due to a technical issue. Participants were aged between 18 and 71 years (mean age 38.2 years), excluding two participants who failed to report their ages. Each participant watched one of 10 short movies and then provided a written recall of the movie plot. The audiovisual movie stimuli were different from the ones used in the fMRI experiment and included both animations and live-action movies. The movies were on average 9.1 min long (ranged 5.9 to 12.7 min). Each movie was watched by 38–49 participants (mean = 39.3). Additional 99 participants were excluded from the analysis because their recall was too short (<150 words) or they had watched the movie before the experiment. All participants were provided with an informed consent form approved by the Johns Hopkins University Institutional Review Board. Participants received monetary compensation for their time ($10 per hour).

The experiment was run in web browsers using JavaScript. After reading instructions, participants watched a 2-min-long example video clip. An example recall of the example video clip was also provided to inform participants about the level of detail they need to produce during recall. Participants then watched a short movie and performed the written recall task by typing in a text box to retell the movie plot in their own words. To mimic the irreversible nature of the spoken recall used in the fMRI experiment, participants were not allowed to backspace beyond the current sentence in order to edit already-written sentences. Participants were encouraged to take as much time as needed to provide as much detail as they can remember. During the delay between movie watching and recall, participants completed a short demographic questionnaire and the Mind-Wandering Questionnaire[86], and then practiced using the text box by providing written

descriptions of simple shapes. Participants also completed a series of questionnaires at the end of the experiment, including the Survey of Autobiographical Memory[87] and the Plymouth Sensory Imagery Questionnaire[88]. Findings from the delay period and post-experiment questionnaires are not reported in this paper.

As in the fMRI experiment, independent coders segmented each movie into discrete events (mean number of events per movie = 25.2) and provided written descriptions of each event. The written recall of each participant was also segmented into sentences, and the movie events that each sentence describes were identified. The semantic and causal narrative networks of the movies were generated using procedures identical to those used in the fMRI experiment, except that 1) the USE vectors were not averaged across annotators as each movie was annotated by a single coder, and 2) a total of 16 independent coders identified the causally related events and each movie was rated by 10 coders.

**Testing statistical assumptions**. To validate the normality assumption for paired *t*-tests comparing the recall probability or mean ROI activation for high vs. low semantic/causal centrality events, we performed Anderson-Darling tests on the differences between conditions. The distributions of the differences did not significantly deviate from a normal distribution (*ps* > .2). For the repeated-measures one-way ANOVAs testing the serial position effects in recall performance, we performed Mauchly's tests of sphericity and confirmed that the assumption of sphericity was not violated (*ps* > .44). We also performed Anderson-Darling tests and confirmed that the distribution of recall probability for each condition did not significantly deviate from a normal distribution (*ps* > .17).

**Citation diversity statement**. Recent work in neuroscience and several other fields of science has identified a citation bias whereby papers from people from underrepresented groups are under-cited[89–93]. Here we aimed to proactively cite references that reflect the diversity of the field in gender, race, and ethnicity. First, we predicted the gender of the first and last author of each reference based on the first names of the authors using online databases[93,94]. Excluding self-citations, the references of the current article contain 8.86% woman(first)/woman(last), 15.53% man/woman, 25.66% woman/man, and 49.95% man/man. Note that this method may not always be indicative of gender identity, and cannot account for intersex, non-binary, or transgender individuals. Similarity, we predicted the race of the first and last author of each reference based on the first and last names of the authors using online databases[95,96]. Excluding self-citations, the references of the current article contain 5.24% author of color (first)/author of color(last), 12.87% White author/author of color, 16.34% author of color/White author, and 65.55% White author/White author. Note that this method may not always be indicative of racial or ethnic identity, and cannot account for Indigenous and multiracial individuals, or those who may face differential biases due to the ambiguous racialization or ethnicization of their names.

**Reporting summary**. Further information on research design is available in the Nature Research Reporting Summary linked to this article.

## Data availability

Source data associated with the figures are provided with this paper. The raw fMRI and behavioral data generated in this study have been deposited in OpenNeuro.org (https://doi.org/10.18112/openneuro.ds004042.v1.0.0)[97]. The region-of-interest labels, activation maps from univariate analysis, movie annotations, and raw behavioural data from the preregistered online experiment are available at GitHub (https://github.com/jchenlab-jhu/filmfest)[98]. Source data are provided with this paper.

## Code availability

The analyses in the current manuscript used code available through MATLAB, R, and Python. Custom scripts that can be used to reproduce the figures from the source data are included in the Source Data file. Other analysis scripts are available upon request to the corresponding author (H.L.).

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

## Acknowledgements

We thank Kenneth A. Norman, Uri Hasson, Christopher J. Honey, Qihong Lu, and Yuan Chang Leong for scientific discussions and comments on earlier versions of the manuscript. We thank Savannah Born and Elly Yeom for their assistance with collecting behavioral ratings. We thank Buddhika Bellana and Yoonjung Lee for their assistance with online experiment preparation and data analysis. This work was supported by the Sloan Research Fellowship to J.C. (FG-2018-10490) and Google Faculty Research Award to J.C. (Google LLC 3-9-2020).

## Author contributions

H.L. and J.C. conceived and designed the experiments. H.L. and J.C. performed the experiments. H.L. analyzed the data. H.L. drafted the paper. H.L. and J.C. edited the paper.

## Competing interests

The authors declare no competing interests.
