## [Peer Review File · Nature Communications]

Predicting memory from the network structure of naturalistic eventsREVIEWER COMMENTS

Reviewer #1 (Remarks to the Author):

Re: Narratives as Networks: Predicting Memory from the Structure of Naturalistic Events

In this study the authors report an fMRI study of event memory, where participants viewed and recalled a series of short films in the scanner. The films themselves were annotated and these annotations were converted into vectors using Google's Universal Sentence Encoder. The authors then used the vectors to investigate the semantic similarity of the sub-events depicted in the films – a "narrative network". The authors report a number of findings: events that are semantically similar to a greater number of other events are typically remembered better, elicit larger responses in the hippocampus at their offset (during encoding), elicit greater activation in default mode network regions of the brain (during recall), and are associated with more similar BOLD patterns of activity across individuals. There are a number of slightly unexpected aspects of the results (e.g. effects of causality between events were not as clear as effects of semantic similarity, few effects of semantic similarity during encoding), but these are thoughtfully addressed in the Discussion.

This is a very solid study – it has been carefully designed and the data are appropriately analysed. It is a shame that the recent paper by Heusser et al., (2021) "Geometric models reveal behavioural and neural signatures of transforming experiences into memories" is conceptually quite similar to the present paper – as this undermines the novelty of the approach and some of the findings. These authors also converted narratives to vectors and report similar findings with respect to the lack of primacy and recency effects as well as correlations with BOLD activity patterns within the default model network. Nevertheless, the similarities between the studies are acknowledged by the authors and the previous study certainly does not diminish the quality of the present one. It's also nice to see that at a conceptual level, the findings are consistent across the studies.

I only have one other minor point. With respect to DMN regions showing the effects, the discussion focuses on the posterior midline cortex. However, there has been a lot of research implicating lateral DMN regions – such as the angular gyrus and middle temporal gyrus in combining semantic information, either at the level of words (Price et al., 2015), or in order to understand sentences (Humphries et al., 2007) or movie scenes (Keidel et al., 2018). The later study showed increases in activation in various lateral DMN regions when participants could link what they were watching with preceding narrative information. Recently, Branzi et al., (2021) used TMS to demonstrate a causal role for the angular gyrus in linking narrative contextual information. Notwithstanding the differences between these studies and the present study (e.g. effects were mostly at encoding, not recall), these studies all seem to add to the weight of evidence that the DMN represents semantically linked information when processing narratives.

Price, A. R., Bonner, M. F., Peelle, J. E., & Grossman, M. (2015). Converging evidence for the neuroanatomic basis of combinatorial semantics in the angular gyrus. *Journal of Neuroscience*, 35(7), 3276-3284.

Humphries, C., Binder, J. R., Medler, D. A., & Liebenthal, E. (2007). Time course of semantic processes during sentence comprehension: an fMRI study. *Neuroimage*, 36(3), 924-932.

Keidel, J. L., Oedekoven, C. S., Tut, A. C., & Bird, C. M. (2018). Multiscale integration of contextual information during a naturalistic task. *Cerebral Cortex*, 28(10), 3531-3539.

Branzi, F. M., Pobric, G., Jung, J., & Lambon Ralph, M. A. (2021). The left angular gyrus is causally involved in context-dependent integration and associative encoding during narrative reading. *Journal of cognitive neuroscience*, 33(6), 1082-1095.

Reviewer #2 (Remarks to the Author):

In the present article, the authors examined how the semantic relatedness and causality between events in narratives influence brain activity during viewing of the narratives and recall of the narratives. The authors observed that events that were more semantically related to other events or shared cause-effect relationships with other events (high centrality events) were recalled more often. During movie viewing, the hippocampus showed a larger event offset for high centrality events. Further, stimulus driven activity fluctuations (intersubject functional connectivity) were correlated between the hippocampus and posterior medial cortex, but not early visual cortex for high centrality events. During recall, activity was higher in posterior medial and lateral parietal cortices and participants had similar multivariate spatial patterns in posterior medial cortex when recalling events that had high semantic centrality compared to low.

The behavioral results reaffirm the rich literature that exists supporting the role of structured knowledge in memory for events. The present study makes a significant methodological contribution by using novel computational techniques by using the Universal Sentence Encoder to create weights for a network that treats events as nodes and semantic relatedness as connections. The imaging findings are novel in showing that activity and representations in the posterior medial cortex during events are related to the interconnectedness of events in a broader narrative.

I would also like to commend the authors on their pre registration and commitment to open and equitable science.

Overall, this paper is innovative and makes an important contribution to the literature. Although centrality might not be the right measure to focus on, the big idea presented here is nonetheless novel and important, and it will stimulate thoughtful work in the field. However, there are several conceptual and analysis issues that need to be addressed before this paper is appropriate for publication.

Conceptual issues

A. The key innovation in this paper is the introduction of centrality as a key measure to explain recall of naturalistic events and fMRI data related to event encoding and retrieval. With this in mind, the paper could do a better job of motivating this particular measure and explaining how this relates to theories of discourse processing. The centrality measure captures sentence-level overlap of events, which is somewhat correlated with, but different from subjective causality judgments. If one draws on discourse theories (e.g., Kintsch) one might think about the extent to which centrality relates to the surface features (e.g. the “textbase”) vs. the situation model vs. the narrative (if there are subplots within the film).

B. Related to the point raised above, it is puzzling why the paper distinguishes between “semantic” and “causal” relationships, as if a causal relationship is not semantically important? Moreover, two events might share sentences that have similar causal relationships (“Janice threw the ball” and later “Chris threw the frisbee”) even when the events themselves are not causally-related and refer to different narratives (baseball game vs disc golf match). My sense is that the centrality measure is relatively insensitive to these nuances, in which case I suggest eliminating this distinction entirely. (Note--most cognitive neuroscientists have no idea what semantics are, let alone causality, etc. so I (CR) strongly encourage the editor to disregard any reviewer criticisms that would say “this is just ‘semantic’ overlap”.

C. Based on the argument presented in the paper, it seems more useful to differentiate event-level information from information at the level of words. For instance, does centrality predict recall over and above word-level overlap? How does centrality in the USE matrix fare against more primitive word-level approaches like word2vec? Other low-level features such as # of words or mean word frequency would be useful to rule out as well.

D. For centrality, the networks constructed did not have directionality. Causes and effects might have different representations in the brain (Leshinskaya et al 2020) and by eliminating the directionality,

these differences could obscure the findings presented in the manuscript. Why was directionality eliminated and what are the implications of a directionless cause-effect network for this study? It would at least be useful to know whether “cause” events differ in recall probabilities as compared to “effect” events.

E. Overall, the discussion section was thin. This is clearly an innovative paper that challenges standard ways of thinking about memory, but the discussion section did not effectively capture the novel contribution of this study.

fMRI analyses:

A. When calculating semantic or causal centrality, the network is informed by all connections. At initial viewing, participants would not have the prospective knowledge of what events are going to subsequently be semantically related to upcoming events (barring the predictable structure some stories might take). In fact, the authors note “While speculative, the diminished effect of centrality on pISC during movie watching may reflect that the structure of the whole narrative becomes apparent only after subjects finished watching the movies (i.e., during recall).” Indeed, it is notable that the correlation between importance ratings as people viewed the movies vs. those taken after the movies is not that impressive ($r=.67$). To account for this issue, centrality could be calculated, for each event, on the relationships between events leading up to and including each given event. This may illuminate the previously null findings found at movie viewing. As noted by the authors, at retrieval this would not be an issue because the participant would have viewed the full movie and thus would have been exposed to the full network of semantic/cause-effect information.

B. The same logic listed above applies to the intersubject functional connectivity analysis, during movie viewing. The relationship between hippocampal activity timecourse and PMC should be more influenced by semantic relationships leading up to and including the current event, but less influenced by future semantic relationships that have yet to be experienced. Centrality measures for a given event should be calculated using only information up to and including that event for this analysis as well. Regarding the findings of this analysis, the authors note “The stronger hippocampal-PMC connectivity during higher centrality events might reflect greater reinstatement of other event representations cued by overlapping components”. This interpretation in particular should be more true of events that occur later in the movies, because future events could not be reinstated in early events since they have not yet been seen and thus, we would not expect to see the hippocampal-PMC connectivity.

C. The authors note that “The hippocampus showed higher activation following the offset of high centrality events, suggesting that stronger hippocampus-mediated encoding contributed to the high centrality advantages.” This suggestion can be explored to some extent by examining whether stronger hippocampal offset activity is associated with successful recall. A mediation analysis could be performed to see if high centrality events are associated with higher hippocampal activity which subsequently predict memory retrieval.

D. If the central claim is that centrality is sensitive to factors that are specific to a particular narrative, it would be good to rule out the possibility that fMRI correlates of centrality are also narrative-specific. In other words, if one computes centrality across movies, rather than within a movie, would this eliminate the effects of centrality on brain activity? Alternatively, it may be the case that high centrality events reflect fairly familiar schemas, one might expect across-movie centrality to also relate to brain activity.

Behavior:

A. The discussion makes the point that no serial position effects were observed in event recall, and although they qualify this argument, it seems that more caution may be warranted. Inspection of Supplemental Fig. 5c suggests that ceiling effects are common for many of the recalled events, possibly obscuring the ability to see serial position effects. Moreover, with naturalistic stimuli that are composed of temporally extended events, it might be difficult to measure serial position effects unless the narrative is sufficiently long to adequately see a dip for events in the middle. Other factors might also be at play, such as the fact that these stimuli seem to have a peculiar linear organization where

each event is judged as being caused by the preceding event. Again, this might blur serial position effects. There are also some more interesting possibilities--for instance, serial position effects might only occur in naturalistic stimuli within an event, but not across events. Finally, this may be an example of Kahneman's famous "peak-end" rule--although I think autobiographical memory research hasn't fully supported Kahneman's story, I believe there is evidence that "highs" and "lows" can be as or more salient than the beginning and end.

B. The present study capitalized on variations in semantic and causal structure within the narratives to look at their influence on behavioral and neural responses. However, given that these factors were not directly manipulated by the authors, it would be important to show that there is good variation in semantic and causal structure across events within each film.

Minor comments

-In the discussion, the authors note that the high centrality events are recalled more frequently than low centrality events. They then go on to say "Consistent with this behavioral effect, higher centrality was associated with greater hippocampal activity at event boundaries, as well as with increased hippocampal-cortical interaction during movie watching." The behavioral response and brain response do not provide any inherent consistency with each other. That is, recall and hippocampal BOLD changes are not measuring the same variable. Rather, these two pieces of evidence can be seen as consistent with a particular theory that makes predictions about behavior and brain activity.

-The events identified in this study seem to be shorter in length than the events often used in studies of memory for film stimuli (e.g., Chen et al., 2017, Nat Neuroscience). This is not a design weakness per se, and indeed it could be a strength. Could the authors elaborate on the reasoning for this design choice and whether shorter time windows for events might affect pattern estimations?

- The paper argues that events with high centrality have a high degree of significance in narrative construction. It would be helpful to give readers a subjective sense of how this plays out in the stimuli--for example, the authors might include in the supplemental section the annotated description (or a summary sentence) of the sequence of events in one or two of the movies, noting whether each event is of high or low centrality. (Also, once this work is published, this information should be made available for all the movies, along with the other data that is to be publicly released)

-The abstract states, "During encoding, central events evoked larger hippocampal event boundary responses associated with memory consolidation" and a similar statement is made in the main text: "This offset response has been interpreted as the registration or consolidation..." It isn't clear why the authors refer to consolidation here, as there isn't any evidence to suggest that the boundary-evoked response is reflecting anything other than (hippocampal) encoding (e.g., Lu, Hasson, & Norman, BIORXIV). It seems unnecessary for the authors to bring in the baggage of consolidation, but if the authors believe this is important, it seems necessary to clarify what they mean (systems? cellular?) and why they believe the effect is related to consolidation per se.

-The authors defined events through event segmentation by an independent coder who "was instructed to identify event boundaries based on major shifts in the narrative (e.g., location, topic, and/or time)." This is a fairly specific way of operationalizing event boundaries, as opposed to the more typical event segmentation approach employed by Zacks and colleagues, which uses more subjective, open-ended instructions. In the Zacks approach, segmentation agreement across different individuals is typically used to define events. I am not saying that the authors need to use Zacks' approach, but it would be useful for the authors to consider (perhaps in the methods section) differences between the two approaches. In particular, Zacks' model emphasizes prediction error as the factor that defines event boundaries, whereas the authors' approach seems to specifically identify points of narrative change. In other words, you might have large prediction errors even when there is no major narrative shift. On a related note, would the "sub-events" identified by the annotators be akin to subjective event segmentation at the coarse level, or are they more fine-grained?

- On the graphs in Figure 2 A, there are three letters. To what are these referring?
- The authors found a relationship between hippocampal-PMC ISFC and centrality but not hippocampus-EVC ISFC and centrality. However, they should test whether there is an interaction effect such that the hypothesized PMC relationship is greater than the control region.

- The authors sometimes refer to “offset” responses and sometimes refer to “boundary” responses, but the distinction (if any) is not explained clearly. I can see the value in discussing boundaries as dividing events and then distinguishing boundaries that define the beginning of an event and boundaries that define the end of an event. Also, this is up to the authors’ discretion, but it might be good to abandon the use of “offset responses”. Although Ben-Yakov and Dudai have used this terminology in previous work, the term seems to imply that there is something particular about the end of a narrative (as is the case in most of their work), as opposed to the end of an event within a narrative.

- The authors present an explanation in the discussion for their hippocampal offset findings stating “one possibility is that the conclusion of a higher centrality event produces greater uncertainty in the ongoing narrative”. It is not abundantly clear to me why this would be the case and the authors should explain why they think this could happen.

- For the RSA recall analysis, the authors averaged the USE matrices constructed by recall transcripts across subjects. Why not keep the subject-specific USE matrix based on each subjects’ individual recall transcript?

- It seems odd that the causal connectivity matrix is driven so heavily by temporal contiguity, but at least some of the movies have considerably more off diagonal causal connections. Do those types of narratives differ from the more linear narratives?

- Multiple raters assessed causality and importance for each events, but we could not find any reliability estimates. It would be important to know whether there was high across-rater reliability in these ratings.

- It appears that some films have subplots, in which case there are multiple narratives, whereas others appear to have a single narrative. Does the interpretation of centrality differ in these two cases?

- It is worth considering whether work on community structure in statistical learning (e.g., Anna Schapiro, Dani Bassett, etc.) is relevant to the approach taken here.

Signed,
Alex Barnett
Charan Ranganath (I sign all reviews)

Reviewer #3 (Remarks to the Author):

Summary

Lee and Chen ran two experiments that had participants watch a sequence of 10 short films and then (verbally or via typed responses) recall what had happened in the films, in any order. One experiment used neuroimaging during movie viewing and recall, and the second (behavior-only) experiment was run on Amazon Mechanical Turk, providing a stronger test of the key behavioral findings. The paper reports several important advances in the study of memory for naturalistic stimuli and experiences. First, the authors build “semantic networks” by applying text embedding models to annotations of each film, providing a clever means of studying how different events are conceptually related. Second, the authors use this network to label events according to their centrality. They find that events with high centrality are better remembered than low-centrality events. Further, hippocampal

responses track with the offsets of high-centrality events, and hippocampal-PMC ISFC is higher during high (vs. low) centrality events. Overall this is an exciting paper, and appropriate for Nature Communications. I have several suggestions, comments, and suggestions for strengthening the paper:

Major comments:

1.) The application of USE to "automatically" identify semantic links between events is clever. I'm also left wondering what specifically leads to high (vs. low) centrality. For example, the authors seem to suggest (based on hand-labeled causality links) that USE-based associations might track with causal associations between events. However, the correlation between causal and semantic centrality is relatively low ($r = 0.28$). I'm wondering if the authors might be able to dig more into the underpinnings of semantic centrality. For example, is the overlap between (semantically) "central" events specific to their videos? E.g., are central events like miniature "summaries" that consolidate (or incorporate information from) many other events in the same video? Or are central events more like non-specific narrative signposts, e.g. that might incorporate common themes that are *not* necessarily video-specific? One way of getting at this would be to compute the similarity between high-centrality events *across* videos, and compare that to the similarities between low-centrality events across videos (and/or similarities between high vs. low centrality events across videos). If high-centrality events were similar to each other in a non-video-specific way, it could suggest they might be playing some sort of general purpose narrative scaffolding role. On the other hand, if high-centrality events seem to overlap only with other events from the same video, that could instead suggest that they are playing a role in linking or consolidating across events within a single narrative. (Either would be interesting!)

2.) I was a bit confused about the "onset" analysis reported in Figure 6b. The "offset" analysis (6a) makes sense to me-- e.g., it shows that the hippocampus might be playing a role in encoding a just-concluded high-centrality event. But what is the motivation driving the onset version of the analysis? For example, how would the participant "know" that they were about to experience a high-centrality event? Some clarification would be useful. Or alternatively, perhaps there is a different way of getting at the question of whether the hippocampus might play a role in encoding event onsets. One possibility would be to look at the *preceding* event's identity as a high-centrality or low-centrality event. For example, is the "offset effect" (6a) stronger when two high-centrality events occur in succession?

Minor comments:

1.) The authors define centrality as normalized degree. However, this might conflate some important event information, and I wonder if this could be resulting in lower correlations with centrality. In particular, suppose that two events (A and B) have the same degree, but event A is connected to higher-centrality events than event B. By the current measure, A and B would be considered equally central. But an alternative measure, such as eigenvector centrality, would reflect that A has more "influence" in the network than B.

2.) The authors might consider describing how centrality was computed in the main text, and possibly even adding a figure with some intuitions about how high vs. low centrality events "look like" in the network (e.g., something like Fig. 1b, but calling out a detail of a high and low centrality event). Although the dot sizes in Figure 1 are useful, it takes quite a bit of zooming in to really appreciate the connections to any single event.

3.) A few of the reported cutoffs for the ISFC seemed arbitrary to me and could benefit from some additional explanation or intuition. For example, the authors first use a 22.5 second (15 TR) cutoff when examining hippocampal-PMC ISFC, and then they report "comparable" results when using a 19.5 second (13 TR) cutoff. Is the analysis sensitive to the precise cutoff? Perhaps it would be cleaner to report results for a *range* of cutoff values? Or to report the range over which the results are "comparable" in some way?

4.) At several points throughout the manuscript, the way the authors report "null" results (e.g., no significant difference between conditions) might imply that there is evidence of no difference (e.g., lines 165--168, 219--220, 398--400). I'd suggest clarifying that no conclusions can be drawn about those results, rather than that they imply a lack of effect per se.

5.) I'd be curious as to how the authors settled on using USE as opposed to other available models. Was the decision somewhat arbitrary (or based on ease of implementation), or was there a deeper theoretical reason for embedding text using USE?

REVIEWER COMMENTS

Reviewer #1 (Remarks to the Author):

In this study the authors report an fMRI study of event memory, where participants viewed and recalled a series of short films in the scanner. The films themselves were annotated and these annotations were converted into vectors using Google's Universal Sentence Encoder. The authors then used the vectors to investigate the semantic similarity of the sub-events depicted in the films – a "narrative network". The authors report a number of findings: events that are semantically similar to a greater number of other events are typically remembered better, elicit larger responses in the hippocampus at their offset (during encoding), elicit greater activation in default mode network regions of the brain (during recall), and are associated with more similar BOLD patterns of activity across individuals. There are a number of slightly unexpected aspects of the results (e.g. effects of causality between events were not as clear as effects of semantic similarity, few effects of semantic similarity during encoding), but these are thoughtfully addressed in the Discussion.

This is a very solid study – it has been carefully designed and the data are appropriately analysed. It is a shame that the recent paper by Heusser et al., (2021) "Geometric models reveal behavioural and neural signatures of transforming experiences into memories" is conceptually quite similar to the present paper – as this undermines the novelty of the approach and some of the findings. These authors also converted narratives to vectors and report similar findings with respect to the lack of primacy and recency effects as well as correlations with BOLD activity patterns within the default model network. Nevertheless, the similarities between the studies are acknowledged by the authors and the previous study certainly does not diminish the quality of the present one. It's also nice to see that at a conceptual level, the findings are consistent across the studies.

: We thank the reviewer for their positive assessment of our study.

I only have one other minor point. With respect to DMN regions showing the effects, the discussion focuses on the posterior midline cortex. However, there has been a lot of research implicating lateral DMN regions – such as the angular gyrus and middle temporal gyrus in combining semantic information, either at the level of words (Price et al., 2015), or in order to understand sentences (Humphries et al., 2007) or movie scenes (Keidel et al., 2018). The later study showed increases in activation in various lateral DMN regions when participants could link what they were watching with preceding narrative information. Recently, Branzi et al., (2021) used TMS to demonstrate a causal role for the angular gyrus in linking narrative contextual information. Notwithstanding the differences between these studies and the present study (e.g. effects were mostly at encoding, not recall), these studies all seem to add to the weight of evidence that the DMN represents semantically linked information when processing narratives.

Price, A. R., Bonner, M. F., Peelle, J. E., & Grossman, M. (2015). Converging evidence for the neuroanatomic basis of combinatorial semantics in the angular gyrus. *Journal of Neuroscience*, 35(7), 3276-3284.

Humphries, C., Binder, J. R., Medler, D. A., & Liebenthal, E. (2007). Time course of semantic processes during sentence comprehension: an fMRI study. *Neuroimage*, 36(3), 924-932.

Keidel, J. L., Oedekoven, C. S., Tut, A. C., & Bird, C. M. (2018). Multiscale integration of contextual information during a naturalistic task. *Cerebral Cortex*, 28(10), 3531-3539.

Branzi, F. M., Pobric, G., Jung, J., & Lambon Ralph, M. A. (2021). The left angular gyrus is causally involved in context-dependent integration and associative encoding during narrative reading. *Journal of cognitive neuroscience*, 33(6), 1082-1095.

: We appreciate the reviewer's suggestion for considering lateral DMN regions in our discussion. We also thank the reviewer for pointing out relevant prior studies. We focused our analyses on the posterior medial cortex because the area showed the strongest event-level representations in the cortex during recall both in our earlier study (Chen et al., 2017) and in the current study. However, as the reviewer anticipated, lateral DMN areas, especially the lateral parietal cortex including the angular gyrus, showed similar effects to PMC with respect to inter-event structure in the whole-brain univariate activation and pattern-based analyses in our study, although the effects were slightly weaker in the lateral DMN areas. We revised a paragraph in the discussion section of the manuscript (below) to emphasize the involvement of the lateral DMN areas, citing some of the papers suggested by the reviewer.

p.18. We demonstrated that DMN activity during remembering was modulated by the recollected event's position in the narrative network. **High-level associative areas in the DMN¹⁴, especially the PMC and its functionally connected subregions such as the angular gyrus, have been implicated in the episodic construction and representation of events^{16,17,53}. In accordance with this view, we observed event-specific neural activation patterns in the medial and lateral DMN areas during recall (Figure 4d), and representational similarity analysis revealed that the relational structure of these neural event patterns could be predicted by human-generated descriptions of the movie and by recall transcripts (Figure 5, Supplementary Figure 12b; for a similar approach, see ref.²⁶). Critically, activation in the PMC and angular gyrus scaled with the degree to which events had more connections with other events during recall (Supplementary Figures 9b, 9d), consistent with prior studies showing that these areas are involved in combining and comprehending semantically connected information⁵⁶⁻⁵⁸.**

Reviewer #2 (Remarks to the Author):

In the present article, the authors examined how the semantic relatedness and causality between events in narratives influence brain activity during viewing of the narratives and recall of the narratives. The authors observed that events that were more semantically related to other events or shared cause-effect relationships with other events (high centrality events) were recalled more often. During movie viewing, the hippocampus showed a larger event offset for high centrality events. Further, stimulus driven activity fluctuations (intersubject functional connectivity) were correlated between the hippocampus and posterior medial cortex, but not early visual cortex for high centrality events. During recall, activity was higher in posterior medial and lateral parietal cortices and participants had similar multivariate spatial patterns in posterior medial cortex when recalling events that had high semantic centrality compared to low.

The behavioral results reaffirm the rich literature that exists supporting the role of structured knowledge in memory for events. The present study makes a significant methodological contribution by using novel computational techniques by using the Universal Sentence Encoder to create weights for a network that treats events as nodes and semantic relatedness as connections. The imaging findings are novel in showing that activity and representations in the posterior medial cortex during events are related to the interconnectedness of events in a broader narrative.

I would also like to commend the authors on their pre registration and commitment to open and equitable science.

Overall, this paper is innovative and makes an important contribution to the literature. Although centrality might not be the right measure to focus on, the big idea presented here is nonetheless novel and important, and it will stimulate thoughtful work in the field. However, there are several conceptual and analysis issues that need to be addressed before this paper is appropriate for publication.

: We thank the reviewers for their thorough consideration of our work, and for their conceptual and analytical suggestions which we have incorporated into the revised manuscript. For example, the suggested mediation analysis with regard to the hippocampal offset responses was successful, and added an exciting new element to the results. We also added several supplementary materials including the effects of causal centrality based on directed networks that distinguished causes and effects.

Conceptual issues

A. The key innovation in this paper is the introduction of centrality as a key measure to explain recall of naturalistic events and fMRI data related to event encoding and retrieval. With this in mind, the paper could do a better job of motivating this particular measure and explaining how this relates to theories of discourse processing. The centrality measure captures sentence-level overlap of events, which is somewhat correlated with, but different from subjective causality judgments. If one draws on discourse theories (e.g., Kintsch) one might think about the extent to

which centrality relates to the surface features (e.g. the “textbase”) vs. the situation model vs. the narrative (if there are subplots within the film).

: Thank you for raising this important point, which we agree should be clarified. In short, our position is that both of our measures, *semantic centrality* and *causality centrality*, relate primarily to the situation model, and less so to the surface features or textbase – with the caveat that some surface features should be expected to reflect aspects of the situation model in naturalistic stimuli.

With respect to semantic centrality: Each event is described by at least one but often more than one sentence, and also described by three different independent annotators. All of these different sentences are incorporated into the USE "semantic" embedding for each event. Thus, the event embedding vectors are to some extent abstracted beyond the surface features (textbase) of the original sentences. That is, while sentence embedding similarity between events measures something analogous to sentence overlap (Kintsch, 1986) or "argument overlap" (Kintsch & van Dijk, 1978), it is also true that the sentence embeddings for a given event are capturing information which spans multiple sentences and phrasing choices. Similarity between event vectors quantifies the connections (edges) within our "semantic" narrative network, which in turn allows calculation of the "semantic centrality" of each event. However, one cannot fully disentangle situation model from surface features in these data. Some surface features should be expected to reflect aspects of the situation model in naturalistic stimuli, e.g., if a dog is present in two events, it is reasonable that annotators will use the word “dog” when describing both events.

With respect to causal centrality, cause-effect relations between events are associated more with the situation model than with the textbase, as exemplified in van Dijk & Kintsch's 1983 definition, wherein "causes and goals" are identified as a crucial elements of knowledge structures; and demonstrated empirically in studies such as Kintsch, 1986; Zwaan et al., 1995. Indeed, half of our videos have no dialogue at all, and thus the human raters cannot be relying on text or language provided in the stimulus to make their judgment. It may be the case that other surface features (auditory or visual) influence their cause-effect ratings in some way; however, their explicit instructions were to identify causal relations between events in the movie narratives, with no mention made of the surface features. Below we provide the exact instructions given to our raters:

Your job is to identify and make a list of event pairs that are causally related to each other within each movie.

How can we decide whether two events are causally related or not? In an extremely broad sense, one might say that any event that happened before a target event could be at least partially responsible for the event to happen (e.g., you were born because there was Big Bang), but this wouldn't give us very useful information. So we want to identify only those event pairs that are more strongly related, and you will need to use your own

best judgment to decide whether the causal relationship is strong enough. For example, if we have a movie like below,

Event 1: Jane orders a crab cake at a restaurant.

Event 2: Jane finds a dead fly in her crab cake.

Event 3: Jane complains to the manager of the restaurant.

You may say that there is a causal relationship between Event 2 and Event 3, but not between Event 1 and Event 3. We don't really have strict rules or criteria, so it is up to your subjective judgment. But please try to keep your criteria as consistent as possible.

Human-judged cause-effect relations between events quantify the connections (edges) within our "causal" narrative network, which in turn allows calculation of the "causal centrality" of each event.

Thus, we would argue that both semantic centrality and causal centrality reflect aspects of how situation models for individual events are related to each other. As the reviewer notes, the two measures are somewhat correlated but also different. This topic is addressed in more detail in our response to the next comment below (Reviewer 2, Conceptual Issues B).

With respect to situation model vs. narrative: the films used in this experiment did not have subplots. However, we would speculate that if a film/story had two largely unrelated subplots, it would be more accurate to separately create narrative networks for the two subplots. In our dataset, there is a somewhat analogous situation in that ten movies were presented consecutively; we chose to create separate networks for each movie, as opposed to entering all movies together in a single giant network. Please see our response to a later comment (Reviewer 2, fMRI analyses D), as well as to Reviewer 3, Major comment 1, for analyses of a giant network incorporating all ten movies.

We have clarified our description of the two types of centrality measures in the Introduction:

pp. 4-5. To quantify and assess the semantic relationship between events within a movie, we employed an approach scalable and easily generalizable to different types of narratives (Figure 1). In this method, each narrative is transformed into a network of interconnected events based on semantic similarity measured from sentence embedding distances (**the "semantic" narrative network**). **We then calculate *semantic centrality* for each event as the node degree, a graph metric which quantifies the number and strength of connections that a node (event) has to other nodes in the network.** Behavioral results revealed that events with higher semantic centrality were more likely to be recalled, without showing primacy and recency effects typical in traditional random list memory experiments^{3,28}. High centrality events were also associated with the neural signatures of stronger and more accurate recall: greater activation and more consistent neural patterns across individuals in the DMN areas including the posterior medial cortex (PMC). The hippocampus showed higher activation following the offset of high centrality

events, suggesting that stronger hippocampus-mediated encoding contributed to the high centrality advantages. **In parallel, we created a “causal” narrative network for each movie based on causal relations between events defined by human judgments. Causal centrality of events, again defined as node degree in the network, predicted memory success and neural responses in a similar way to semantic centrality,** but also made an independent contribution to each.

In the Discussion, we have added a new paragraph to relate our findings to theories of discourse processing and elaborate the distinction between situation models and surface features:

p.20. What is the nature of information reflected in narrative network centrality? We believe that both semantic centrality and causal centrality primarily reflect aspects of how situation models for individual events are related to each other, rather than surface features (or "textbase") of the stimuli or annotations as discussed in classic discourse theories^{70,71}. With respect to semantic centrality, each event annotation included descriptions from three different annotators and often consisted of multiple sentences. All of these different sentences and phrasing choices were incorporated into the text embedding for each event, from which the semantic narrative network arises. Thus, the event embedding vectors capture information abstracted beyond the surface features of the original sentences. With respect to causal centrality, cause-effect relations between events are traditionally associated with the situation model, identified as crucial elements of knowledge structures^{70,72,73}. Indeed, half of the movies in the current study contain no dialogue at all, and thus human raters cannot be relying on text or language provided in the stimulus to make causality judgments. Thus, while some surface features such as annotators' word choices should be expected to reflect aspects of the situation model (see Supplementary Methods for semantic centrality based on word-level information), situation-level information, rather than low-level textual overlap, is likely to determine our centrality measures.

We have also added the causality judgment instructions shown above to Supplementary Methods.

B. Related to the point raised above, it is puzzling why the paper distinguishes between “semantic” and “causal” relationships, as if a causal relationship is not semantically important? Moreover, two events might share sentences that have similar causal relationships (“Janice threw the ball” and later “Chris threw the frisbee”) even when the events themselves are not causally-related and refer to different narratives (baseball game vs disc golf match). My sense is that the centrality measure is relatively insensitive to these nuances, in which case I suggest eliminating this distinction entirely. (Note--most cognitive neuroscientists have no idea what semantics are, let alone causality, etc. so I (CR) strongly encourage the editor to disregard any reviewer criticisms that would say “this is just ‘semantic’ overlap”).

: We thank the reviewers for raising this important question. We completely agree that “semantic” and “causal” relationships within a narrative are closely related; our motivation for including both in this paper was that we wanted to assess the relationship “strength” between event pairs, and neither one of our measures (semantic or causal) alone appeared to fully capture this relationship. Indeed, each seemed to be missing something crucial. Text embedding models are not designed or trained to assess whether two events are causally related. Meanwhile, human judgments of causality do not necessarily depend on “semantic similarity” as assessed by text embeddings; e.g., two events “Janice threw the ball” and “Chris fell to the ground, unconscious” may have a clear causal link for a reader with some background knowledge, but low semantic similarity according to text embeddings (no overlapping or similar-meaning words or topics).

We did find that semantic centrality and causal centrality metrics were correlated:

p.9. Causal centrality was positively correlated with semantic centrality ($r(202) = .28, p < .001, 95\% \text{ CI} = [.15, .41]$).

But we also found that they were not redundant:

p.9. A mixed-effects logistic regression analysis revealed that semantic centrality explains successful event recall even after controlling for causal centrality ($\beta = .17, \text{ standard error (SE)} = .05, \chi^2(1) = 12.24, p < 0.001$) and vice versa ($\beta = .38, \text{ SE} = .05, \chi^2(1) = 55.04, p < 0.001$).

These results were replicated in our pre-registered online experiment:

pp.9-10. We conducted a pre-registered online experiment ($N = 393$) and replicated the same behavioral characteristics of narrative recall using a new set of 10 short movies (Supplementary Figure 7). Each subject watched one of the movies and then performed a free written recall of the movie plot. Consistent with the behavioral results from the fMRI experiment, semantic centrality ($\beta = .17, \text{ SE} = .03, \chi^2(1) = 48.52, p < 0.001$) and causal centrality ($\beta = .44, \text{ SE} = .03, \chi^2(1) = 255.67, p < 0.001$) each uniquely predicted the successful recall of an event

We suspect that further investigations with more stories (beyond the twenty in this paper) would reveal that the two frequently co-occur in naturalistic narratives. However, given that there are clear logical and statistically-supported distinctions between the two measures, at least as operationalized in this paper, we deemed it more conservative to report the two sets of results separately.

We now expand on the co-occurrence and distinction between the two types of relationships in the discussion section:

pp.20-21. Causal relations have long been considered an important organizing factor for

event and narrative memories^{22,65,74}. **Consistent with earlier work, we found that events with stronger causal connections with other events are better remembered (Figures 3c-d), and these effects were not redundant to those of semantic connections.** Yet, while the effects of causality on univariate responses during movie-viewing (see also ref.⁶⁴) were comparable to the effects of semantic centrality (Supplementary Figures 9c, 10c), multivoxel pattern effects of causality during recall were not as clear as those of semantic similarity (Supplementary Figures 10b, e). Several characteristics of causal relations in movie stimuli might have reduced the reliability of the effects of causal narrative network structure. First, causal relations were sparse and mostly identified between adjacent events (Supplementary Figures 4a, 5d). In addition, causality judgments may be more idiosyncratic: average across-coder correlation was lower for causal (mean $r(202) = .34$) than semantic centrality (mean $r(202) = .52$) when centrality was computed from each individual coder's causality rating or movie annotation. **It is also noteworthy that semantic and causal connections were measured in distinct ways (text embeddings and human judgments, respectively) and reflect different types of information:** semantic connections are based on similar or shared features such as people, places, and objects, whereas causal connections additionally require an action, its outcome, and internal models providing a logical dependency between the two^{75,76}. **For example, two events “Jill threw the ball” and “Jack fell to the ground, unconscious” may have a clear causal link for a reader with some background knowledge, but low semantic similarity according to text embeddings as they have no overlapping or similar-meaning words or topics. In this study, we did not focus strongly on dissociating semantic and causal centrality, as they were positively correlated in our movie stimuli.** Future studies designed to orthogonalize different types of inter-event relations, including semantic and causal relations as well as other dimensions such as emotional similarity^{77,78}, will be able to further clarify their unique influences on the behavioral and neural signatures of memory. **Additionally, further investigations with more stimuli may examine the extent to which the two co-occur in naturalistic narratives, as well as in non-narrative real-world experiences.**

Also, to err on the side of caution, we would like to address a possible confusion in the reviewer's example. The reviewer wrote:

“two events might share sentences that have similar causal relationships (“Janice threw the ball” and later “Chris threw the frisbee”) even when the events themselves are not causally-related and refer to different narratives (baseball game vs disc golf match). My sense is that the centrality measure is relatively insensitive to these nuances...”

Please note that the “causal relationships” we measured in our study only involved the causality *between* events; no causal structure present in a sentence *within* an event affected causality judgments.

pp.25-26. independent coders identify causally related event pairs (the 'cause' event and the 'effect' event) within each movie. ...an event pair always consisted of two different events.

In other words, these two events “Janice threw the ball” and “Chris threw the frisbee” would be scored as having *high relatedness* in our “semantic” measure, but these same two events could plausibly be scored as *unrelated* according to our “causal” measure (if indeed one does not cause the other). The two different centrality measures would reflect these two different scores, and thus the centrality measures do indeed have the capacity to be sensitive to such nuances.

C. Based on the argument presented in the paper, it seems more useful to differentiate event-level information from information at the level of words. For instance, does centrality predict recall over and above word-level overlap? How does centrality in the USE matrix fare against more primitive word-level approaches like word2vec? Other low-level features such as # of words or mean word frequency would be useful to rule out as well.

: The reviewers raise an interesting question about centrality effects on recall—whether event-level information can be differentiated from word-level information, with a progression of levels suggested: from the low-level features such as word counts/frequencies, to the higher levels of word-level overlap and word-based text embeddings such as word2vec.

Starting at the bottom, we examined whether the number of words per event (i.e., in the annotation of any given movie event) could be used to predict semantic centrality. We found no significant relationship ($r = .1061, p = .13$). We also examined another “low-level” feature correlated with the number of words, event duration; this feature also had no significant relationship with centrality ($r = .1091, p = .12$). We next examined whether the mean word frequency of movie event descriptions could be used to predict semantic centrality. We used the word frequency database downloaded from <https://www.wordfrequency.info/samples.asp>. We found no significant relationship between the mean word frequency and centrality ($r = .027, p = .7$). Thus, it is unlikely that these low-level features, which do not incorporate word meanings, were driving the effects of the sentence embedding-based semantic centrality.

At a higher level of information suggested by the reviewers was word overlap between events. We deem this “higher level” because it incorporates information about word meanings and thus semantic similarity between events. We measured the word overlap (exact matching words) between events as the Jaccard index for event pairs within each annotator and movie. Then the Jaccard indices were averaged across annotators. Not surprisingly, the semantic centrality computed from the networks based on the word-level overlap (whose edge weights were the mean Jaccard indices) was positively correlated with semantic centrality computed from USE embeddings ($r(202) = .64, p < .001, 95\% \text{ CI} = [.55, .71]$). This type of semantic centrality was also correlated with recall probability ($r(202) = .27, p < .001, 95\% \text{ CI} = [.14, .39]$).

Word-based text embeddings models such as word2vec also convey higher-level information. While these models do technically make predictions at the level of individual words, it is not

really accurate to think of them as carrying information *restricted* to individual words. Such models are typically trained using context windows of several words on either side (skip-gram or CBOW approach), and thus the embedding vector for a single word necessarily incorporates information from the surrounding context in which it is found in the training corpus – indeed, this is the very reason why word embedding models perform so well. (E.g., a word model trained on scrambled text corpora would be a failure.)

Thus, USE embedding vectors corresponding to a given sentence are closely related to the average of word embeddings corresponding to the words in the same sentence. Given this, it is unsurprising that we find that the two methods give rise to very similar narrative network structure. Event-wise semantic centrality generated via the two methods (USE and word2vec) was positively correlated ($r(202) = .54$, $p < .001$, 95% CI = [.43, .63]). Word2vec-based semantic centrality was correlated with recall probability ($r(202) = .34$, $p < .001$, 95% CI = [.21, .46]), as the USE-based semantic centrality was.

In summary, event centrality calculated from low-level features (number of words, event duration, and mean word frequency) was not significantly correlated with sentence embedding-based centrality. Event centrality calculated via event-level averaging of individual word-level overlap or word embeddings produced similar results to sentence-level embeddings, and these approaches are all valid ways of defining semantic similarity between events (as is the related method of topic modeling, e.g., Heusser et al. 2021). We chose USE over word-level overlap or word2vec for its simplicity, because USE takes full sentences as input; using word-level overlap, word2vec, or other word-embeddings methods requires additional steps and decisions (e.g., whether to apply text-cleaning steps such as stemming and selecting a list of non-content “stop words” to remove), as well as additional computations (e.g., averaging across many word embeddings for each event), involving choices for which there is no principled field standard. Using USE allowed us to minimize preprocessing of the annotation texts and preserve the natural descriptions of the human annotators as much as possible.

Another relevant point is that semantic centrality computed from USE embeddings was more consistent across different annotators (mean cross-annotator similarity $r(202) = .53$) compared to that of word-level overlap ($r(202) = .42$) or word2vec embeddings ($r(202) = .50$). This may be due to the fact that annotators often used very different sets of words to describe the same event. For example, below are two different annotators’ descriptions of an event in the movie *Catch Me If You Can*:

Annotator 1: Camera cuts to a prison hallway. Inmates are up against a chain link fence, rattling it ringing bells and making noise.

Annotator 2: The scene opens to prisoners banging metal objects against a wire fence and yelling.

Although it is clear that the two descriptions share much in terms of semantic content, there is only one word, “fence”, that overlaps across the two annotators. Thus, the multidimensional projections of the sentence-level information are likely to be more robust and reliable measure

of semantic similarity between events compared to methods that depend on word-level matching.

We have added the results of the above analyses to the text:

p.25. *Semantic narrative networks*. Movie annotations were used to generate narrative networks based on the semantic similarity between events (Figure 1). For each annotator and movie, the text descriptions for the fine-grained sub-events were concatenated within each movie event. The text descriptions were then encoded into high-dimensional vectors with Google's Universal Sentence Encoder (USE²⁵) such that each movie event was represented as a 512-dimensional vector. The USE vectors from the three annotators were highly similar to each other (mean event-wise cross-annotator cosine similarity between all possible annotator pairs = .78; Supplementary Figure 1); thus the USE vectors were averaged across annotators within each movie event. For each movie, the narrative network was generated by using the cosine similarity between the USE vectors of movie event pairs as the edge weights between nodes (events). **The semantic centrality values based on USE sentence embedding vectors were correlated with those based on word-level overlap or word2vec embeddings (Supplementary Methods).**

Supplementary Methods pp.1-2. *Semantic narrative networks based on word-level information*.

To test the effects of semantic centrality based on word-level rather than sentence-level similarity between movie event annotations, we created two additional types of semantic narrative networks. First, we created narrative networks whose edge weights between events were defined as Jaccard indices reflecting the word overlap (exact matching words) between event text descriptions. The Jaccard indices were computed within each annotator, and then averaged across annotators within each movie. As in the USE-based narrative networks, event centrality was defined as the normalized node degree. We found that the semantic centrality computed from the networks based on Jaccard indices was positively correlated with the semantic centrality based on USE embeddings ($r(202) = .64, p < .001, 95\% \text{ CI} = [.55, .71]$) and also with recall probability ($r(202) = .27, p < .001, 95\% \text{ CI} = [.14, .39]$). Second, we created networks whose edge weights between events were the cosine similarity between the word embeddings of the events. Specifically, the word embedding of an event was generated by averaging the word vectors (based on Google's pre-trained Word2Vec model; GoogleNews-vectors-negative300-SLIM) of unique words contained in the text description of the event, separately for each annotator. The word embeddings were then averaged across annotators. Words that were not included in the Google's database were excluded from the analysis. The centrality (normalized node degree) computed from these networks was again positively correlated with the USE-based semantic centrality ($r(202) = .54, p < .001, 95\% \text{ CI} = [.43, .63]$) and with recall probability ($r(202) = .34, p < .001, 95\% \text{ CI} = [.21, .46]$).

D. For centrality, the networks constructed did not have directionality. Causes and effects might have different representations in the brain (Leshinskaya et al 2020) and by eliminating the directionality, these differences could obscure the findings presented in the manuscript. Why was directionality eliminated and what are the implications of a directionless cause-effect network for this study? It would at least be useful to know whether “cause” events differ in recall probabilities as compared to “effect” events.

: We thank the reviewers for highlighting this topic. In our initial approach to the analysis, we also wanted to incorporate directionality to the causal network. However, several factors led us to abandon it for the final analyses.

First, we discovered that the movie stimuli in our experiment had mostly linear causal structure; a movie event is often both the cause of its upcoming event and the effect of its preceding event. Thus, “cause” events and “effect” events are likely to have similar recall probabilities in our study.

Second, we wanted to directly compare the effects of semantic centrality and causal centrality, and this comparison is more interpretable if both semantic and causal networks are undirected networks (semantic networks are undirected by necessity of their definition).

Third, we realized that centrality based only on either causes or effects in directed networks would be correlated with the centrality computed from undirected networks and produce similar effects. This is because the degree centrality of a node is identical to the sum of the outdegree and the indegree of the node.

To address the reviewers’ question, we generated causal networks considering the directionality of causes and effects. As expected, outdegree centrality (causes) and indegree centrality (effects) were both highly positively correlated with the centrality computed from undirected casual networks (outdegree: $r(202) = .79, p < .001, 95\% \text{ CI} = [.73, .83]$; indegree: $r(202) = .71, p < .001, 95\% \text{ CI} = [.63, .77]$). Moreover, both outdegree and indegree centrality predicted recall behaviors in the same way as the centrality computed from undirected causal networks did. Outdegree centrality was positively correlated with recall probability ($r(202) = .22, p = .002, 95\% \text{ CI} = [.09, .35]$), and the top 40% outdegree centrality events were better remembered than the bottom 40% outdegree centrality events ($t(14) = 5.9, p < .001, \text{Cohen’s } d_z = 1.52, 95\% \text{ CI of difference} = [.05, .11]$). Likewise, indegree centrality was positively correlated with recall probability ($r(202) = .24, p < .001, 95\% \text{ CI} = [.1, .36]$), and the top 40% indegree centrality events were better remembered than the bottom 40% indegree centrality events ($t(14) = 7.34, p < .001, \text{Cohen’s } d_z = 1.89, 95\% \text{ CI of difference} = [.07, .13]$). We now report these results in Supplementary Figure 6 and in the method section of the revised manuscript as below:

p.26. For each movie, the edge weights between nodes in the narrative network was defined as the proportion of coders who identified a movie event pair as causally related, regardless of the cause-effect direction. **However, causal centrality computed from directed networks which accounted for the cause-effect direction showed highly**

similar behavioral effects as the centrality computed from undirected networks (Supplementary Figure 6).

Supplementary Figure 6. Relationship between recall performance and causal centrality computed from directed networks. We generated directed causal narrative networks where source nodes were “cause” events and target nodes were “effect” events. The edge weight of a cause-effect event pair was defined as the proportion of coders who identified the pair as causally related. An event has high outdegree centrality if the event causes many other events. An event has high indegree centrality if the event is caused by many other events. Both outdegree and indegree centrality were both positively correlated with the centrality computed from undirected casual narrative networks (outdegree: $r(202) = .79, p < .001, 95\% \text{ CI} = [.73, .83]$; indegree: $r(202) = .71, p < .001, 95\% \text{ CI} = [.63, .77]$). **a.** Correlation between outdegree centrality and recall probability. **b.** Recall probability for High (top 40%) vs. Low (bottom 40%) outdegree centrality events defined within each movie (averaged across movies). **c.** Correlation between indegree centrality and recall probability. **d.** Recall probability for High (top 40%) vs. Low (bottom 40%) indegree centrality events defined within each movie (averaged across movies). In **a** and **c**, each dot represents an individual movie event. Different colors denote different movies. In **b** and **d**, white circles represent individual subjects. Black diamonds represent the mean across subjects within each condition. Error bars show SEM across subjects. $**p < .01, ***p < .001$.

E. Overall, the discussion section was thin. This is clearly an innovative paper that challenges standard ways of thinking about memory, but the discussion section did not effectively capture the novel contribution of this study.

: We believe that the discussion section has been significantly strengthened by the sections we added in the course of carefully addressing all four reviewers’ comments. Below are some of the added/revised paragraphs:

pp.16-17. Recent years have seen an explosion in the use of naturalistic stimuli such as movies and narratives in exploring the behavior and neuroscience of human memory, as they provide an engaging laboratory experience with strong ecological validity compared to isolated words or pictures^{7,8,44}. These studies have suggested that findings from traditional random-item list paradigms, which have dominated the field for decades, do not always fully extend to naturalistic recall (e.g., ref.²⁶). In line with this, we observed that the recall probability of events from a movie does not show serial position effects typically reported in random-item list learning^{3,28} where the first and last few items in a list tend to be better remembered than items in the middle. This finding was consistent regardless of whether each subject watched a single movie (Supplementary Figure 7c) or a series of movies in a row (Figure 2d). **The lack of clear primacy or recency advantages may be due to the inter-event dependencies which made each narrative a coherent structure, supporting memories for central events which did not necessarily occur at the beginning or end of the story; that is, inter-event connections may overshadow the existing effects of temporal positions.**

p. 18. We demonstrated that DMN activity during remembering was modulated by the recollected event's position in the narrative network. **High-level associative areas in the DMN¹⁴, especially the PMC and its functionally connected subregions such as the angular gyrus, have been implicated in the episodic construction and representation of events^{16,17,49}. In accordance with this view, we observed event-specific neural activation patterns in the medial and lateral DMN areas during recall (Figure 4d),** and representational similarity analysis revealed that the relational structure of these neural event patterns could be predicted by human-generated descriptions of the movie and by recall transcripts (Figure 5, Supplementary Figure 12b; for a similar approach, see ref.²⁶). **Critically, activation in the PMC and angular gyrus scaled with the degree to which events had more connections with other events during recall (Supplementary Figures 9b, 9d), consistent with prior studies showing that these areas are involved in combining and comprehending semantically connected information⁵²⁻⁵⁴.**

Furthermore, higher semantic centrality predicted greater between-subject pattern convergence in PMC (Figure 4e). This is likely to be a neural signature of stronger and more accurate recall of episodic details^{17,55} for high centrality events, dovetailing with the behavioral results. Additionally, higher intersubject similarity for high centrality events might arise from “design pressure” on narratives. Highly connected events are likely to be logically important in a story; indeed, we found that semantic centrality was positively correlated with the perceived importance of events as retrospectively rated by independent coders ($r(202) = .22, p = .002, 95\% \text{ CI} = [.08, .34]$). Thus, to aid the understanding of their linked events and eventually the whole story, high centrality events need to be designed in a way that minimizes the variability or ambiguity in how people interpret them. This adoption of a similar canonical interpretation of an event across people gives rise to more similar neural responses across individuals⁵⁶⁻⁵⁸. **The design pressure may even produce unique characteristics associated with high centrality shared across events and narratives; high centrality events were**

semantically more similar to other high centrality events than to low centrality events across movies, although the difference was small (difference in $r = .047$, $p < .001$). Future work may investigate whether real-life everyday events without such design pressure would show similar centrality effects to what we observed here using fictional narratives.

p.20. **What is the nature of information reflected in narrative network centrality? We believe that both semantic centrality and causal centrality primarily reflect aspects of how situation models for individual events are related to each other, rather than surface features (or "textbase") of the stimuli or annotations as discussed in classic discourse theories^{70,71}. With respect to semantic centrality, each event annotation included descriptions from three different annotators and often consisted of multiple sentences. All of these different sentences and phrasing choices were incorporated into the text embedding for each event, from which the semantic narrative network arises. Thus, the event embedding vectors capture information abstracted beyond the surface features of the original sentences. With respect to causal centrality, cause-effect relations between events are traditionally associated with the situation model, identified as crucial elements of knowledge structures^{70,72,73}. Indeed, half of the movies in the current study contain no dialogue at all, and thus human raters cannot be relying on text or language provided in the stimulus to make causality judgments. Thus, while some surface features such as annotators' word choices should be expected to reflect aspects of the situation model (see Supplementary Methods for semantic centrality based on word-level information), situation-level information, rather than low-level textual overlap, is likely to determine our centrality measures.**

pp.20-21. Causal relations have long been considered an important organizing factor for event and narrative memories^{22,65,74}. **Consistent with earlier work, we found that events with stronger causal connections with other events are better remembered (Figures 3c-d), and these effects were not redundant to those of semantic connections.** Yet, while the effects of causality on univariate responses during movie-viewing (see also ref.⁶⁴) were comparable to the effects of semantic centrality (Supplementary Figures 9c, 10c), multivoxel pattern effects of causality during recall were not as clear as those of semantic similarity (Supplementary Figures 10b, e). Several characteristics of causal relations in movie stimuli might have reduced the reliability of the effects of causal narrative network structure. First, causal relations were sparse and mostly identified between adjacent events (Supplementary Figures 4a, 5d). In addition, causality judgments may be more idiosyncratic: average across-coder correlation was lower for causal (mean $r(202) = .34$) than semantic centrality (mean $r(202) = .52$) when centrality was computed from each individual coder's causality rating or movie annotation. **It is also noteworthy that semantic and causal connections were measured in distinct ways (text embeddings and human judgments, respectively) and reflect different types of information:** semantic connections are based on similar or shared features such as people, places, and objects, whereas

causal connections additionally require an action, its outcome, and internal models providing a logical dependency between the two^{75,76}. **For example, two events “Jill threw the ball” and “Jack fell to the ground, unconscious” may have a clear causal link for a reader with some background knowledge, but low semantic similarity according to text embeddings as they have no overlapping or similar-meaning words or topics. In this study, we did not focus strongly on dissociating semantic and causal centrality, as they were positively correlated in our movie stimuli.** Future studies designed to orthogonalize different types of inter-event relations, including semantic and causal relations as well as other dimensions such as emotional similarity^{77,78}, will be able to further clarify their unique influences on the behavioral and neural signatures of memory. **Additionally, further investigations with more stimuli may examine the extent to which the two co-occur in naturalistic narratives, as well as in non-narrative real-world experiences.**

We are currently slightly over the word count limit of an article for *Nature Communications*. If the reviewers would like to suggest a specific extra topic or issue which could further improve the overall quality of the manuscript, we would be happy to consider adding it to the manuscript given the editor’s permission to exceed the word count limit.

fMRI analyses:

A. When calculating semantic or causal centrality, the network is informed by all connections. At initial viewing, participants would not have the prospective knowledge of what events are going to subsequently be semantically related to upcoming events (barring the predictable structure some stories might take). In fact, the authors note “While speculative, the diminished effect of centrality on pISC during movie watching may reflect that the structure of the whole narrative becomes apparent only after subjects finished watching the movies (i.e., during recall).” Indeed, it is notable that the correlation between importance ratings as people viewed the movies vs. those taken after the movies is not that impressive ($r=.67$). To account for this issue, centrality could be calculated, for each event, on the relationships between events leading up to and including each given event. This may illuminate the previously null findings found at movie viewing. As noted by the authors, at retrieval this would not be an issue because the participant would have viewed the full movie and thus would have been exposed to the full network of semantic/cause-effect information.

: We thank the reviewers for suggesting this interesting analysis idea. We had explored computing centrality using only the events leading up to and including each given event as the reviewers suggested, but instead chose to use all events in our final analyses for several reasons.

First, computing node centrality in networks with a very small number of nodes may distort the centrality values, produce unreliable results, or require us to exclude a substantial number of movie events from analyses. For example, centrality simply cannot be computed for the first events of the movies, because there is only one node in the network. The normalized degree

centrality is also always the same value for all second events of the movies (computed from a network with two nodes and one edge).

Second, using the same centrality metrics computed from the same set of networks makes it easier and more justifiable to make comparisons between the movie watching phase results and recall phase results.

Third, centrality computed using only the events leading up to and including each given event is still expected to be similar to centrality computed using all events, especially later in the movies, as more and more events are included in the networks. Below we show the centrality computed from the partial networks:

And below is the centrality computed from the full networks:

It is clear from these two plots that the centrality values of the first few events of each movie are very different between the partial and full networks, but later events have similar values across the two methods. Indeed, there was a strong positive correlation between centrality computed from events leading up to and including each given event and centrality computed using all events ($r = .7556, p < .001$). Thus, employing the 'partial network' approach while excluding the unreliable first few events and using later events in the analyses would be expected to produce very similar results as simply using centrality computed from all events. We now report the positive correlation between the two types of semantic centrality in the discussion section as below:

pp.19. One might have expected that the effects of narrative structure would not be apparent in brain responses measured during ongoing movie watching, as the full structure of inter-event connections is only available after all movie events are completed. Still, as discussed above, event centrality significantly influenced

hippocampal and cortical univariate responses during movie watching. A possible explanation for these results is that centrality based on partial narrative networks (i.e., a network that excluded events not-yet-presented) was sufficiently similar to the full-narrative centrality values, especially later in a movie. **Indeed, semantic centrality computed from networks excluding not-yet-presented events was positively correlated with that based on full networks ($r(192) = .76, p < .001, 95\% \text{ CI} = [.69, .81]$).**

In addition, although the reviewers found it unimpressive, the positive correlation ($r = .67, p < .001$) between importance ratings as people viewed the movies vs. those taken after the movies suggests that people can anticipate the relative importance/centrality of an event within the complete story. This makes it plausible that events presented earlier in a movie could be affected by the full-network centrality.

However, to fully address the reviewers' concern, we performed the intersubject pattern correlation (pISC) analysis again, using the centrality computed from events leading up to and including each given event. Specifically, we compared the movie watching phase pISC between high centrality (top 40%) and low centrality (bottom 40%) events. We used all events other than the first event of each movie (which did not have centrality values). Consistent with the original results we reported in the manuscript, we found that the mean pISC was numerically higher in high than low centrality events in the PMC and numerically higher in low than high centrality events in the early visual cortex, but neither effect was statistically significant (PMC: pISC difference = .0141, randomization test $p = .3047$; early visual cortex: pISC difference = -.0407, randomization test $p = .0509$). Thus, centrality computed using only the events leading up to and including each given event produced similar effects as centrality computed using all events. We have opted not to report these in the revised manuscript. However, we would be happy to include them if the reviewer feels they would be of interest to readers.

B. The same logic listed above applies to the intersubject functional connectivity analysis, during movie viewing. The relationship between hippocampal activity timecourse and PMC should be more influenced by semantic relationships leading up to and including the current event, but less influenced by future semantic relationships that have yet to be experienced. Centrality measures for a given event should be calculated using only information up to and including that event for this analysis as well. Regarding the findings of this analysis, the authors note "The stronger hippocampal-PMC connectivity during higher centrality events might reflect greater reinstatement of other event representations cued by overlapping components". This interpretation in particular should be more true of events that occur later in the movies, because future events could not be reinstated in early events since they have not yet been seen and thus, we would not expect to see the hippocampal-PMC connectivity.

: As the reviewers suggested, we performed the hippocampus-cortex intersubject functional connectivity (ISFC) analysis again using semantic centrality calculated using only information up to and including that event. Consistent with the findings that we reported in response to Comment A above, the new analysis produced qualitatively identical results as the original

analysis using centrality computed from all events within each movie. Specifically, in the 26 movie events which were 22.5 seconds (15 TRs) or longer, there was a positive correlation between the centrality based on partial networks and the hippocampus-PMC ISFC ($r = .4198$, $p = .0328$). In contrast, no significant correlation was found between centrality and the hippocampus-early visual cortex ISFC ($r = -.0072$, $p = .9723$). Again, we have opted not to report these results in the revised manuscript as they are redundant with the original results, but we would be happy to include them if necessary. We also kept the semantic centrality based on full networks to avoid potential issues due to computing centrality from networks consisting of a very small number of nodes and also to make it more justifiable to compare the movie watching phase centrality effects and recall phase centrality effects by using the same set of networks and centrality metrics. Finally, as for testing whether the centrality-ISFC relationship is stronger for events that occur later in the movies, unfortunately the analysis is not feasible in the current study due to the limited number of events within each condition (early vs. late in the movie) that are long enough to compute ISFC (minimum 15 TRs). We reported the minimum duration of events and the number of events included in the ISFC analysis in Supplementary Table 5.

Supplementary Table 5. Relationship between semantic centrality and hippocampal-cortical intersubject functional connectivity (ISFC) during movie watching.

Minimum event duration threshold (sec)	Number of events	Correlation between semantic centrality and hippocampus-PMC ISFC (a)			Correlation between semantic centrality and hippocampus-EVC ISFC (b)			95% CI ² of (a) - (b)
		r	p	95% CI ¹	r	p	95% CI ¹	
27	14	.61	.02	[.11, .86]	-.33	.24	[-.73, .24]	[.1, 1.46]
25.5	16	.59	.02	[.14, .84]	-.32	.23	[-.7, .21]	[.12, 1.42]
24	19	.52	.02	[.09, .79]	-.17	.49	[-.58, .31]	[.01, 1.21]
22.5	26	.49	.01	[.13, .74]	.01	.95	[-.38, .4]	[.04, .87]
21	31	.38	.04	[.02, .64]	.02	.93	[-.34, .37]	[-.04, .72]
19.5	44	.29	.06	[-.01, .54]	.05	.75	[-.25, .34]	[-.07, .53]
18	55	.21	.12	[-.05, .45]	-.02	.88	[-.28, .25]	[-.08, .52]

¹ Confidence Interval of the correlation coefficient, [lower bound, upper bound].

² Confidence interval of the difference between two overlapping correlations based on dependent groups, computed using the method described in ref.⁴².

C. The authors note that “The hippocampus showed higher activation following the offset of high centrality events, suggesting that stronger hippocampus-mediated encoding contributed to the high centrality advantages.” This suggestion can be explored to some extent by examining whether stronger hippocampal offset activity is associated with successful recall. A mediation analysis could be performed to see if high centrality events are associated with higher hippocampal activity which subsequently predict memory retrieval.

: We thank the reviewers for suggesting this insightful analysis. We performed the mediation analysis using R’s “mediation” and “lme4” packages. We found that event-by-event hippocampal offset responses predicted subsequent event recall ($\beta = .26$, $SE = .1$, $\chi^2(1) = 6.37$, $p = 0.012$). The hippocampal event offset responses also mediated the effect of semantic centrality on subsequent event recall success (average causal mediation effects = $.001$, $p = .016$, 95% CI = $[-.0002, 0.003]$). Yet, the effect of semantic centrality was still significant after controlling for the hippocampal responses ($\beta = .2$, $SE = .05$, $\chi^2(1) = 13.91$, $p < 0.001$), indicating a partial mediation. We now report these results in the revised manuscript as below:

pp.14-15. We tested whether the centrality of events influences the offset-triggered hippocampal encoding signal during movie watching, potentially mediating the behavioral effect of narrative network centrality. We measured the time courses of hippocampal BOLD activation locked to the boundaries between events, and found that hippocampal responses were higher following the offset of high than low semantic centrality events (Figure 6a). In contrast, hippocampal responses following the onset of high vs. low centrality events (i.e., before the events fully unfold and diverge in terms of their semantic contents) were not significantly different from each other (Figure 6b), confirming that semantic centrality specifically affected the encoding of information accumulated during just-concluded events. **Stronger hippocampal event offset responses (averaged across 10 – 13 TRs from each offset) also predicted the successful recall of individual events in a mixed-effects logistic regression analysis ($\beta = .26$, $SE = .1$, $\chi^2(1) = 6.37$, $p = 0.012$), consistent with prior studies^{36,37}. Moreover, hippocampal offset responses significantly mediated the effects of semantic centrality on event recall (average causal mediation effects = $.001$, $p = .016$, 95% CI = $[-.0002, 0.003]$); the effect of semantic centrality was still significant after controlling for hippocampal responses ($\beta = .2$, $SE = .05$, $\chi^2(1) = 13.91$, $p < 0.001$), indicating a partial mediation.** These results suggest that rich connections between events lead to stronger hippocampus-mediated encoding.

We also revised the methods section to create a separate “Hippocampal event boundary responses” subsection and report detailed mediation analysis methods as below.

pp.31-32. **Hippocampal event boundary responses**

We compared hippocampal event boundary responses following the onset/offset of high vs. low centrality events during movie watching (Figure 6). High and low centrality events were defined as the events whose centrality values were within the top or bottom 40% in each movie. We first averaged TR-by-TR BOLD signals across voxels within the bilateral hippocampus mask for each subject. We then extracted time series around the onset/offset (-2 – 15 TRs) of each high/low centrality event. The first and last events of each movie were excluded to minimize the effect of between-movie transitions. Each time series was baseline corrected by subtracting the mean activation of the two TRs immediately preceding the onset/offset of the event from each time point. The subject-specific time series were then averaged across events within each condition and then across movies. Two-tailed paired *t*-tests were used for each time point to compare the high vs. low centrality conditions. We applied the Benjamini-Hochberg procedure ($q < .05$) to correct for multiple comparisons across time points.

To test whether the effect of semantic centrality on event-by-event recall success (1 = recalled, 0 = not recalled) was mediated via the hippocampal event offset responses, we performed a mediation analysis. Each event from each subject served as a data point, and data were concatenated across all subjects. For each subject, the hippocampal offset response of each event was computed by averaging the BOLD time series measured from 10 to 13 TRs after the event offset. Again, the responses were baseline corrected for each event by subtracting the mean response of the two TRs immediately preceding the event offset from the time series. The first/last events of each movie and not recalled events were excluded from the analysis. Three mixed-effects linear or logistic regression models were defined to test 1) the total effect of semantic centrality on recall success (logistic), 2) the effect of semantic centrality on hippocampal offset responses (linear), and 3) the direct effect of semantic centrality on recall success, controlling for hippocampal offset responses (logistic). An additional mixed-effects logistic regression analysis was also performed to test the effect of hippocampal offset responses on recall success. In all models, subjects were included as random effects. The significance of the indirect effect of hippocampal offset responses on the relationship between semantic centrality and recall success was tested via the quasi-Bayesian Monte Carlo simulation as implemented in the “mediation” package in R. Specifically, 1000 simulations were performed to compute the 95% confidence interval of the average causal mediation effects.

Finally, we revised the text in the discussion section considering the new results as below:

p.17. The benefit of high centrality during encoding is also reflected in the greater hippocampal responses following the offset of high than low centrality movie events (Figure 6a). **Such hippocampal event boundary responses have been linked to the successful registration of just-concluded episodes into long-term memory^{12,13,35}, which was replicated in the current study.**

D. If the central claim is that centrality is sensitive to factors that are specific to a particular narrative, it would be good to rule out the possibility that fMRI correlates of centrality are also

narrative-specific. In other words, if one computes centrality across movies, rather than within a movie, would this eliminate the effects of centrality on brain activity? Alternatively, it may be the case that high centrality events reflect fairly familiar schemas, one might expect across-movie centrality to also relate to brain activity.

: We agree with the reviewers that it would be useful to test whether the characteristics of high centrality events are shared across different narratives. Reviewer 3 raised a similar question, although about semantic rather than neural characteristics of high centrality events. In our response to Reviewer 3, we found that the USE vectors of high centrality events within a movie are slightly more similar to those of high than low centrality events in other movies, although the effect was small (difference in $r = .047$) and needs to be interpreted with caution.

In response to the current reviewers' question: To test whether the fMRI correlates of centrality are also shared across movies, we performed the analysis that the reviewers suggested. We generated a single giant narrative network including events from all 10 movies and the events are connected only between different movies (i.e., there was no within-movie connection). We then computed the centrality of each event from the giant network. Note that this analysis applies only to semantic centrality, because we assume that human raters would judge there to be zero causal connections between different movies in this stimulus set.

One difficulty with this analysis is that event centrality in the giant network may be heavily influenced by the number of events within each movie and the general similarity between different movies. That is, if movie A and movie B happen to be semantically similar due to having similar themes/topics, and these movies have relatively more events compared to other movies, the events within movies A and B would have overall greater centrality than events within other movies. This indeed seems to be the case, as shown in the plot below (some movies have much higher mean centrality than that of other movies). This bias may distort the centrality effects on neural responses.

Nonetheless, to fully address the reviewers' question, we tested centrality effects on the intersubject pattern similarity (pISC) in PMC, using the centrality calculated from only between-movie connections. We compared the high (top 40%) and low (bottom 40%) semantic centrality events and found that there was no significant difference between the centrality conditions (high centrality mean pISC = .0685, low centrality mean pISC = .0526, randomization test $p = .0919$). We also compared the hippocampal event offset responses for the high vs. low centrality events

and again found no difference between the two conditions, as shown in the plot below (solid line = high centrality, dotted line = low centrality).

These results might suggest that although there may be subtle semantic differences between high and low centrality events shared across different movies, the effects are not strong enough to significantly drive neural differences. However, as we discussed above, there were biases introduced when centrality metrics were computed from between-movie connections, and thus we opted not to report these results in the revised manuscript.

Behavior:

A. The discussion makes the point that no serial position effects were observed in event recall, and although they qualify this argument, it seems that more caution may be warranted.

Inspection of Supplemental Fig. 5c suggests that ceiling effects are common for many of the recalled events, possibly obscuring the ability to see serial position effects. Moreover, with naturalistic stimuli that are composed of temporally extended events, it might be difficult to measure serial position effects unless the narrative is sufficiently long to adequately see a dip for events in the middle. Other factors might also be at play, such as the fact that these stimuli seem to have a peculiar linear organization where each event is judged as being caused by the preceding event. Again, this might blur serial position effects. There are also some more interesting possibilities--for instance, serial position effects might only occur in naturalistic stimuli within an event, but not across events. Finally, this may be an example of Kahneman's famous "peak-end" rule--although I think autobiographical memory research hasn't fully supported Kahneman's story, I believe there is evidence that "highs" and "lows" can be as or more salient than the beginning and end.

: We appreciate the reviewers' thoughts on potential factors that might have obscured existing serial position effects in narratives in our study. However, we believe that our results that showed the lack of serial position effects are robust and real for several reasons.

First of all, the ceiling effect in recall probability for many movie events shown in Supplemental Fig. 5c (7c in the revised manuscript) actually strengthens our findings. The figure clearly shows that many movie events presented in the middle of each movie showed ceiling effects, meaning that memories for those events were excellent even though they were not close to the beginning

or end of the movies, whereas the recall probability for the very first or last movie events was often much lower--which is the opposite of what serial position effects would predict.

Second, the lack of serial position effects in a narrative stimulus was also observed in a much longer movie (Heusser et al., 2021). We believe that the movie was long enough (50 minutes with 50 events) to show a dip for events in the middle. Our results replicated this prior result.

Third, regarding the linear organization of causally related events, there is reason to believe that the linear causal relationship is actually not peculiar --it appeared in all 20 movies used in our study (including the online experiment) and also was shown in prior studies (Trabasso & Sperry, 1985; Song et al., 2021). The causal chain seems to be the normative organization scheme of common narratives. More importantly, if the lack of serial position effects was indeed affected by this causal chain as the reviewers suggest, it directly supports, rather than weakens, our argument that inter-event connections (that are not present in traditional random list stimuli) contribute to the distinct behavioral characteristics of narrative memory. That is, perhaps serial position effects are present, but heavily overshadowed by the memory effects of causal relations.

Fourth, the possibility that serial position effects might be present within an event (but not across events) depends heavily on how 'events' are defined. Specifically, depending on how coarse- or fine-grained events are, individual events in one study can be considered as within-event segments in another study. As the reviewers pointed out in one of their minor comments, our events in the current study were much shorter than those defined in our previous study (Chen et al., 2017) because the movies used in the current study were much shorter. Thus, if we had used the same range of event durations in the current study as in Chen et al. (2017), the events defined in the current study could have become within-event segment units. However, in both the current and the earlier study (relevant behavioral effects reported in Heusser et al., 2021), no serial position effect was observed regardless. Thus, it is very much likely that as long as within-event segments are interrelated, those relationships between segments would have a greater influence on recall than serial positions.

Finally, regarding Kahneman's peak-end rule, we agree that serial position effects may be diluted because of salient highs/lows in the middle, which is in line with our findings.

Nonetheless, we agree with the reviewers about the possibility that serial position effects were present but were not able to be detected in the current study. Thus, we added the following text to our discussion of serial position effects in the paper:

pp.16-17. In line with this, we observed that the recall probability of events from a movie does not show serial position effects typically reported in random-item list learning^{3,28} where the first and last few items in a list tend to be better remembered than items in the middle. This finding was consistent regardless of whether each subject watched a single movie (Supplementary Figure 7c) or a series of movies in a row (Figure 2d). The lack of clear primacy or recency advantages may be due to the inter-event dependencies which

made each narrative a coherent structure, supporting memories for central events which did not necessarily occur at the beginning or end of the story; **that is, inter-event connections may overshadow the existing effects of temporal positions.**

B. The present study capitalized on variations in semantic and causal structure within the narratives to look at their influence on behavioral and neural responses. However, given that these factors were not directly manipulated by the authors, it would be important to show that there is good variation in semantic and causal structure across events within each film.

: We agree with the reviewers that variations in semantic and causal structure within each movie narrative are crucial in our study. Because the aim of our study was to examine the effect of narrative structure in as naturalistic a setting as possible, we did not use an approach of creating artificial narratives with experimental manipulations embedded in them. Instead, we selected a variety of existing commercial movies of different artistic styles and from diverse sources. We showed the variations in narrative network structure in Figure 1b and Supplementary Figures 4a-b (former Supplementary Figures 3a-b), and also showed the variations in semantic and causal centrality across events within each narrative in Figure 1c and Supplementary Figure 4c (former Supplementary Figure 3c). We have now added a new supplementary figure (Supplementary Figure 3, shown below) to show the semantic network structure of all ten movies used in the fMRI experiment (also related to Reviewer 2's minor comment 2). Testing whether the variation in semantic and causal structure across events in our movie stimuli was "good enough" is challenging, as there is no proper criterion; indeed even if these factors had been directly manipulated by us, it is not clear how to test whether there is "good variation". Fortunately, there was enough variance in event-specific semantic and causal centrality to demonstrate their statistically significant effects on behavioral and neural responses (i.e., high centrality benefits in both memory recall performance and fMRI responses in the hippocampus and higher cortical areas).

Supplementary Figure 3. Semantic narrative networks of all movie stimuli. **a.** Semantic similarity matrices of the 10 movies used in the fMRI experiment. **b.** Semantic narrative networks of the 10 movies used in the fMRI experiment. Node size is proportional to centrality (normalized degree) computed from unthresholded networks. Edge thickness is proportional to edge weights. Nodes with brighter colors indicate high (i.e., within the top 40% in each movie) semantic centrality events.

Minor comments

-In the discussion, the authors note that the high centrality events are recalled more frequently than low centrality events. They then go on to say “Consistent with this behavioral effect, higher centrality was associated with greater hippocampal activity at event boundaries, as well as with increased hippocampal-cortical interaction during movie watching.” The behavioral response and brain response do not provide any inherent consistency with each other. That is, recall and hippocampal BOLD changes are not measuring the same variable. Rather, these two pieces of evidence can be seen as consistent with a particular theory that makes predictions about behavior and brain activity.

: Thank you for this comment. We revised the text in the discussion section as below:

p.16. Subjects watched and recounted the movies in their own words; events highly connected with other events within the narrative network, i.e., “high centrality” events, were more likely to be recalled. **Higher centrality was also associated with greater hippocampal activity at event boundaries, as well as with increased hippocampal-cortical interaction during movie watching.**

-The events identified in this study seem to be shorter in length than the events often used in studies of memory for film stimuli (e.g., Chen et al., 2017, Nat Neuroscience). This is not a design weakness per se, and indeed it could be a strength. Could the authors elaborate on the reasoning for this design choice and whether shorter time windows for events might affect pattern estimations?

: The reviewers are correct that the movie events identified in the current study were shorter than the events defined in Chen et al. (2017). The decision to use relatively shorter or finer-grained events (on average 13.3 seconds long) was based on several reasons. First of all, each of our movie stimuli was approximately 2 to 8 minutes long, which was much shorter than the Sherlock movie we used in our earlier study (50 minutes long). Thus, we needed to segment each movie into finer-grained events so that there were enough numbers of events per movie to show its complex narrative network structure. In addition, an earlier study on event segmentation (Zacks et al., 2009) reported the mean perceived “coarse” event duration to be 10 - 20 seconds, although the stimuli used were verbal narratives (not audiovisual movies). Similarly, a recent study (Geerligs et al., 2021) reported that the median “state durations” measured based on neural activation patterns were between 7 to 25 seconds, which was shorter on average than the ~1 minute event duration used in Chen et al. (2017). Thus, we think that using relatively shorter time windows for events was more appropriate for capturing event-specific patterns in short movies used in the current study. We revised the methods section of the manuscript as below to provide rationale for identifying relatively shorter time windows for events in our analyses:

p.23. Following the method used in our previous study¹⁷, we instructed the coder to identify event boundaries based on major shifts in the narrative (e.g., location, topic, and/or time). Unlike in the prior study that used a 50-minute movie¹⁷, we did not set the minimum event duration (10 seconds) because we used much shorter movie stimuli in the current study.

*Zacks, J. M., Speer, N. K., & Reynolds, J. R. (2009). Segmentation in reading and film comprehension. *Journal of Experimental Psychology: General*, 138(2), 307–327.
<https://doi.org/10.1037/a0015305>

*Geerligs, L., Gerven, M. van, Campbell, K. L., & Güçlü, U. (2021). A nested cortical hierarchy of neural states underlies event segmentation in the human brain. *bioRxiv*,
<https://doi.org/10.1101/2021.02.05.429165>

- The paper argues that events with high centrality have a high degree of significance in narrative construction. It would be helpful to give readers a subjective sense of how this plays out in the stimuli--for example, the authors might include in the supplemental section the annotated description (or a summary sentence) of the sequence of events in one or two of the movies, noting whether each event is of high or low centrality. (Also, once this work is published, this information should be made available for all the movies, along with the other data that is to be publicly released)

: We appreciate the reviewers' suggestion for showing an example movie annotation as supplemental information. We added Supplementary Table 3 to show the annotation of the movie "The Record" by an example annotator, and provided the ranks of the 14 events within the movie according to their semantic centrality and casual centrality. We will also make our annotation data public along with the recall transcripts upon the publication of the current manuscript.

Supplementary Table 3. An example movie annotation of "The Record" with event-wise ranks based on semantic centrality and causal centrality (annotation by the annotator RC).

Semantic centrality rank	Causal centrality rank	Event description
14	10	The camera pans into an animated scene with a teenage girl in a room in a tall building. The camera is inside the apartment, and the girl opens the pizza box and grabs a pizza.
13	9	The girl, before she can eat, hears a knock on the door. The girl opens the door and looks around to see no one is there.
9	7	The girl looks down and sees a package for her in an envelope at her doorstep. The girl goes back to her chair and to her pizza and opens the envelope. The girl pulls out a package with a disc inside it that says "A Single Life."
12	7	The girl pulls out the record disc from the package and the title "A film by: Job, Joris, and Marieke"
4	6	The girl gets up and puts the record disc into the record player and puts down the needle. The song on the disc plays and she sits down and begins to eat her pizza.
6	5	The girl is about to eat the pizza but then there is a flash. She stops and then the song plays its lyrics on the song. Part of the pizza is gone.
3	7	She notices the pizza is eaten and then looks at the disc. The camera pans to the disc playing on the record player, and the disc is spinning on the player.
2	1	The girl stops the disc and there's a record scratch. The girl pulls the disc back and forth on the player and pizza disappears and reappears as she tests it back and forth. The girl pulls the disc forth and the pizza pie disappears completely in the box as well as in her hand. Then she pulls it back and the pizza reappears.
1	6	The girl realizes her power and gets up then lifts the needle on the record player. The disc plays the song and then the flash goes to her as a pregnant woman. The woman stops the record player and stops the song from playing.

10	4	The woman pulls the disc forward and back and sees the baby develop and devolve like the pizza before. The woman pulls the disc forward and the baby develops and pops into her arms but the baby starts crying.
5	4	The woman then flashes into her childhood self and looks at herself. The girl goes up to record player and tries to stop it but pops off the needle.
7	3	The girl flashes to her in a wheelchair as an elderly woman and she looks at herself. The woman rolls up her wheelchair but she flashes back to the same scene over and over again, getting frustrated. The woman rolls up again and flashes but then stops rolling up and finds that nothing happens. She then tries to roll really fast but falls back.
8	2	The woman gets up and she is an old woman who needs a walker and is wearing glasses. The woman sees that the song is about to end and tries to get to the record player.
11	8	The woman turns into ashes in a pot in the same nursery home and the record player stops with the needle lifting.

-The abstract states, “During encoding, central events evoked larger hippocampal event boundary responses associated with memory consolidation” and a similar statement is made in the main text: “This offset response has been interpreted as the registration or consolidation...” It isn’t clear why the authors refer to consolidation here, as there isn’t any evidence to suggest that the boundary-evoked response is reflecting anything other than (hippocampal) encoding (e.g., Lu, Hasson, & Norman, BIORXIV). It seems unnecessary for the authors to bring in the baggage of consolidation, but if the authors believe this is important, it seems necessary to clarify what they mean (systems? cellular?) and why they believe the effect is related to consolidation per se.

: We referred to consolidation because prior studies (e.g., Ben-Yakov & Dudai, 2011) reporting the relationship between hippocampal boundary responses and subsequent memory performance associated the results with hippocampal consolidation. However, we agree with the reviewers that it is unnecessary to use the term “consolidation” to describe the hippocampal boundary response result and its interpretation in the current manuscript. We revised the text in several sections of the manuscript not to include the term “consolidation” as below:

p.2. During encoding, central events evoked larger hippocampal event boundary responses associated with **memory formation**.

p.14. This boundary response has been interpreted as **the registration of the just-concluded event into long-term memory**.

p.17. Such hippocampal event boundary responses have been linked to **the successful registration** of just-concluded episodes into long-term memory^{12,13,35}, which was replicated in the current study.

-The authors defined events through event segmentation by an independent coder who “was instructed to identify event boundaries based on major shifts in the narrative (e.g., location, topic, and/or time).” This is a fairly specific way of operationalizing event boundaries, as opposed to the more typical event segmentation approach employed by Zacks and

colleagues, which uses more subjective, open-ended instructions. In the Zacks approach, segmentation agreement across different individuals is typically used to define events. I am not saying that the authors need to use Zacks' approach, but it would be useful for the authors to consider (perhaps in the methods section) differences between the two approaches. In particular, Zacks' model emphasizes prediction error as the factor that defines event boundaries, whereas the authors' approach seems to specifically identify points of narrative change. In other words, you might have large prediction errors even when there is no major narrative shift. On a related note, would the "sub-events" identified by the annotators be akin to subjective event segmentation at the coarse level, or are they more fine-grained?

: As the reviewers commented, our event segmentation instructions asked the coder to identify the moments of situation transitions, and the instructions were more specific compared to instructions used in Zacks and colleagues' studies ("press a button to identify the largest/smallest units of activity that were natural and meaningful to them"). However, we believe that overall both methods would result in similar event boundaries, especially when the boundaries are salient (i.e., there is a greater agreement across coders in Zacks' approach). This is because both segmentation methods are ultimately based on similar information, as the 'prediction error' that defines event boundaries generally arises when there are changes in the current situation (e.g., Huff, M., Meitz, T. G. K., & Papenmeier, F., 2014, JEP:LMC). Thus, we opted not to emphasize the differences between the two methods in the manuscript, as it would not necessarily clarify our methods or findings. We decided to use the current event segmentation method following our prior study (Chen et al., 2017) which resulted in event boundaries that produced reasonable behavioral and fMRI results. We thus edited the "Movie event segmentation" subsection of Methods as below to clarify that we chose our method of event segmentation based on our prior study:

p.23. Following the method used in our previous study¹⁷, we instructed the coder to identify event boundaries based on major shifts in the narrative (e.g., location, topic, and/or time).

The sub-events identified by the annotators were more fine-grained than the 202 events mainly used in the analyses, as each of the 202 events was further segmented into multiple sub-events. This is specified in the "Movie annotations" subsection of Methods in the manuscript as below:

p.23. Each annotator identified finer-grained sub-event boundaries within each of the 202 movie events based on their subjective judgments.

-On the graphs in Figure 2 A, there are three letters. To what are these referring?

: The letters indicate subject-unique IDs. As shown in revised Figure 2 below, we replaced the letters with "Subject A" and "Subject B" to make it clear that they refer to different subjects.

-The authors found a relationship between hippocampal-PMC ISFC and centrality but not hippocampus-EVC ISFC and centrality. However, they should test whether there is an interaction effect such that the hypothesized PMC relationship is greater than the control region.

: We apologize for the ambiguity in reporting our results. To test whether the correlation between hippocampus-PMC ISFC and centrality is greater than the correlation between hippocampus-EVC ISFC and centrality, we computed Zou’s (2007) confidence interval of the difference between two overlapping correlations based on dependent groups. The 95% confidence interval was [.05, .87], suggesting that the two correlations were significantly different (i.e., the confidence interval does not include zero). We now report this result in the text as below:

pp.15-16. In contrast, the hippocampal-EVC interaction did not show a significant relationship with centrality ($r(26) = .01, p = .95, 95\% \text{ CI} = [-.38, .4]$), and the correlation was significantly lower than that between hippocampus-PMC ISFC and centrality (95% CI of the difference between correlations⁴² = [.05, .87]).

*Zou, G. Y. (2007). Toward using confidence intervals to compare correlations. *Psychological Methods*, 12, 399-413. doi:10.1037/1082-989X.12.4.399

-The authors sometimes refer to “offset” responses and sometimes refer to “boundary” responses, but the distinction (if any) is not explained clearly. I can see the value in discussing boundaries as dividing events and then distinguishing boundaries that define the beginning of an event and boundaries that define the end of an event. Also, this is up to the authors’ discretion, but it might be good to abandon the use of “offset responses”. Although Ben-Yakov and Dudai have used this terminology in previous work, the term seems to imply that there is something particular about the end of a narrative (as is the case in most of their work), as opposed to the end of an event within a narrative.

: We revised the text as below to replace the term “offset response” with “boundary response” as suggested by the reviewers. The term “offset response” was used only once in the manuscript.

p.14. **This boundary response** has been interpreted as the registration of the just-concluded event into long-term memory.

-The authors present an explanation in the discussion for their hippocampal offset findings stating “one possibility is that the conclusion of a higher centrality event produces greater uncertainty in the ongoing narrative”. It is not abundantly clear to me why this would be the case and the authors should explain why they think this could happen.

: We appreciate the reviewers’ comment and the opportunity to clarify our interpretation of the greater hippocampal responses following the offset of high centrality events. Our speculation was (Discussion p.19), that “highly connected events are likely to be logically important in a story; indeed, we found that semantic centrality was positively correlated with the perceived importance of events as retrospectively rated by independent coders ($r(202) = .22, p = .002, 95\% \text{ CI} = [.08, .34]$).” That is, the conclusion of a logically significant higher centrality event may have a greater influence on the flow of the ongoing narrative, which can produce greater uncertainty. We edited the following paragraph in the discussion section to clarify this:

pp.17-18. The benefit of high centrality during encoding is also reflected in the greater hippocampal responses following the offset of high than low centrality movie events (Figure 6a). Such hippocampal event boundary responses have been linked to the successful registration of just-concluded episodes into long-term memory^{12,13,35}, which was replicated in the current study. It has been shown that DMN connectivity during movie-viewing is modulated by surprise⁵¹; **one possibility is that the conclusion of a higher centrality event produces greater uncertainty in the ongoing narrative, as higher centrality events are more likely to influence the main storyline of the narrative. This may result in a more salient boundary and stronger boundary-evoked encoding signals.**

-For the RSA recall analysis, the authors averaged the USE matrices constructed by recall transcripts across subjects. Why not keep the subject-specific USE matrix based on each subjects' individual recall transcript?

: We used the single recall USE matrix, averaged across subjects, to make the analysis more comparable to the movie watching phase RSA where we used a single text-based similarity matrix based on the movie annotations. We performed the recall RSA analysis using USE matrices generated from each subject's own recall transcript, and found similar results: parcels within the default mode network, especially the posterior medial cortex, showed the strongest representational similarity between the recall transcripts and fMRI data. The maps below show the parcels with significantly positive representational similarity after multiple comparisons correction.

As this result is largely redundant with the original result using the averaged USE matrix (except that the effects were relatively weaker probably due to the greater influence of idiosyncrasy in subjects' recall), we have opted not to include this analysis in the manuscript. However, we would be happy to include the RSA map in Supplementary Figure 12 if the reviewers think that it will be of interest to readers.

-It seems odd that the causal connectivity matrix is driven so heavily by temporal contiguity, but at least some of the movies have considerably more off diagonal causal connections. Do those types of narratives differ from the more linear narratives?

: The reviewers' insight is correct that most of the causal relationships our coders identified are between temporally adjacent events (as shown in Supplementary Figures 5c and 5d the revised manuscript). Similar results have been reported in a recent study also using a naturalistic movie viewing paradigm (Song, Park, Park, & Shim, 2021, *JNeuro*), suggesting that it might be a common property of the causal structure of commercial movies. It is also true that some of the movies have relatively more complex causal structures than others, but we did not find any qualitative differences in terms of the effect of causal centrality between those movies and

“simpler” ones. Below we show the causal connectivity matrix and the recall probability of the high vs. low causal centrality events for each of the ten movies used in the fMRI experiment. As you may see, the memory benefit of high causal centrality was observed in all but one movie, and there is no obvious relationship to the relative complexity of a story’s causal structure.

In addition, for the pre-registered online experiment, we deliberately selected movie stimuli with more complex structures (i.e., more off-diagonal causal connections) compared to the ones used in the fMRI experiment. Below we show the ten movies’ causal connectivity matrices and recall probabilities for high vs. low centrality. Again, all but one movie stimuli showed memory benefits for high causal centrality.

-Multiple raters assessed causality and importance for each events, but we could not find any reliability estimates. It would be important to know whether there was high across-rater reliability in these ratings.

: We thank the reviewers for pointing out the important missing information. We now report the reliability across raters for importance ratings and causality judgments in the methods section of the manuscript as below:

p. 24. These rate-as-you-go importance ratings averaged across the raters were positively correlated with the retrospective ratings ($r(202) = .67, p < .001, 95\% \text{ CI} = [.58, .74]$). **Importance ratings were positively correlated across raters for both retrospective ratings and rate-as-you-go ratings (mean event-wise cross-rater correlation computed within each movie = .65 and .55, respectively).**

p. 26. For each movie, the edge weights between nodes in the narrative network were defined as the proportion of coders who identified a movie event pair as causally related, regardless of the cause-effect direction. However, causal centrality computed from directed networks which accounted for the cause-effect direction showed highly similar behavioral effects as the centrality computed from undirected networks (Supplementary Figure 6). **The average Jaccard similarity between a pair of coders' lists of causally**

related event pairs was .31 (computed within each movie and then averaged across movies).

-It appears that some films have subplots, in which case there are multiple narratives, whereas others appear to have a single narrative. Does the interpretation of centrality differ in these two cases?

: Our movie stimuli all had a single narrative “thread” per movie; there was no movie stimulus containing subplots with distinct storylines. We speculate that the high centrality benefit would be observed within each subplot in stories consisting of very different subplots, as we observed the centrality effect within each movie.

-It is worth considering whether work on community structure in statistical learning (e.g., Anna Schapiro, Dani Bassett, etc.) is relevant to the approach taken here.

: We thank the reviewers for directing our attention to potentially relevant prior studies. The statistical learning literature and our study use very different types of stimuli (simple isolated stimuli vs. complex and continuous narratives) and also have different emphasis; the statistical learning studies focus on the learning of network structure through the transition probability between repeating elements (nodes), whereas the current study focuses on the effect of the existing/given network structure on the formation of one-shot episodic memories for individual nodes. Yet, we agree that the studies raised by the reviewers are highly relevant, and thus we added the following text in the Discussion to acknowledge the prior work and provide a link to the current study:

p. 19. Future work will explore how brain responses are driven by the temporally evolving, rather than static, inter-event structure when subjects consume unpredictable stories, or actively engage in selecting upcoming narrative events. **Future work may also explore the cognitive and neural mechanisms supporting the learning of novel narrative network structures, and whether they are similar to learning the network structure of simple isolated stimuli or actions (e.g., refs.^{68,69}).**

Signed,
Alex Barnett
Charan Ranganath (I sign all reviews)

Reviewer #3 (Remarks to the Author):

Summary

Lee and Chen ran two experiments that had participants watch a sequence of 10 short films and then (verbally or via typed responses) recall what had happened in the films, in any order. One experiment used neuroimaging during movie viewing and recall, and the second (behavior-only)

experiment was run on Amazon Mechanical Turk, providing a stronger test of the key behavioral findings. The paper reports several important advances in the study of memory for naturalistic stimuli and experiences. First, the authors build "semantic networks" by applying text embedding models to annotations of each film, providing a clever means of studying how different events are conceptually related. Second, the authors use this network to label events according to their centrality. They find that events with high centrality are better remembered than low-centrality events. Further, hippocampal responses track with the offsets of high-centrality events, and hippocampal-PMC ISFC is higher during high (vs. low) centrality events. Overall this is an exciting paper, and appropriate for Nature Communications. I have several suggestions, comments, and suggestions for strengthening the paper:

: We thank the reviewer for their enthusiasm and positive assessment of our work.

Major comments:

*1.) The application of USE to "automatically" identify semantic links between events is clever. I'm also left wondering what specifically leads to high (vs. low) centrality. For example, the authors seem to suggest (based on hand-labeled causality links) that USE-based associations might track with causal associations between events. However, the correlation between causal and semantic centrality is relatively low ($r = 0.28$). I'm wondering if the authors might be able to dig more into the underpinnings of semantic centrality. For example, is the overlap between (semantically) "central" events specific to their videos? E.g., are central events like miniature "summaries" that consolidate (or incorporate information from) many other events in the same video? Or are central events more like non-specific narrative signposts, e.g. that might incorporate common themes that are *not* necessarily video-specific? One way of getting at this would be to compute the similarity between high-centrality events *across* videos, and compare that to the similarities between low-centrality events across videos (and/or similarities between high vs. low centrality events across videos). If high-centrality events were similar to each other in a non-video-specific way, it could suggest they might be playing some sort of general purpose narrative scaffolding role. On the other hand, if high-centrality events seem to overlap only with other events from the same video, that could instead suggest that they are playing a role in linking or consolidating across events within a single narrative. (Either would be interesting!)*

: We thank the reviewer for suggesting an interesting analysis idea that allows us to more thoroughly explore the nature of USE-based semantic centrality. We performed the analysis that the reviewer suggested to test whether the USE vectors for high centrality events in one movie were more similar to those of high or low centrality events in other movies. We found that on average, the correlation between high centrality events was higher than the correlation between high and low centrality events. The difference was small but significant (difference in $r = .047$, $p < .001$) compared against the null distribution generated by randomly shuffling the high and low centrality event labels within each movie. Thus, it seems like high centrality events share some characteristics generalizable across movies. However, the shared characteristics are likely to be subtle and not easy to interpret, as different movies had clearly distinct contents in our experiment. We revised the discussion section of the manuscript as below to address this result:

p.18. Highly connected events are likely to be logically important in a story; indeed, we found that semantic centrality was positively correlated with the perceived importance of events as retrospectively rated by independent coders ($r(202) = .22, p = .002, 95\% \text{ CI} = [.08, .34]$). Thus, to aid the understanding of their linked events and eventually the whole story, high centrality events need to be designed in a way that minimizes the variability or ambiguity in how people interpret them. This adoption of a similar canonical interpretation of an event across people gives rise to more similar neural responses across individuals⁵⁶⁻⁵⁸. **The design pressure may even produce unique characteristics associated with high centrality shared across events and narratives; high centrality events were semantically more similar to other high centrality events than to low centrality events across movies, although the difference was small (difference in $r = .047, p < .001$).** Future work may investigate whether real-life everyday events without such design pressure would show similar centrality effects to what we observed here using fictional narratives.

We also added a subsection in the methods section to provide the details of the analysis:

p.26. **Semantic similarity between events across movies**

To examine whether there are semantic characteristics shared among high semantic centrality events across different movies, we computed similarity between event-specific USE vectors (averaged across annotators) across movies. Specifically, we tested whether the similarity between high centrality events was higher than the similarity between high and low centrality events. High and low centrality events were defined as the events whose semantic centrality values were within the top and bottom 40% in each movie, respectively. For each movie, we computed Pearson correlations between the USE vector of each high centrality event and the USE vectors of each of the other movies' high centrality events. The correlation coefficients were averaged across events and movies to produce the mean similarity value for high centrality-high centrality event pairs. Likewise, we computed the mean similarity between each movie's high centrality events and each of the other movies' low centrality events. We then performed a randomization test to assess whether the difference between the mean similarities of high-high pairs and high-low pairs was significantly different from zero. A null distribution of the difference of mean USE vector similarities was generated by randomly shuffling the high or low centrality labels of the events within each movie and then computing the difference 1000 times. A two-tailed p -value was defined as the proportion of values from the null distribution equal to or more extreme than the actual difference.

In addition, we added an extra supplementary table (Supplementary Table 3) to show text descriptions (annotations) for individual events within an example movie and the events' ranks based on semantic/causal centrality values (also related to a minor question of Reviewer 2). We believe that this will further help readers understand the nature of high and low centrality events in the movie stimuli.

Supplementary Table 3. An example movie annotation of “The Record” with event-wise ranks based on semantic centrality and causal centrality (annotation by the annotator RC).

Semantic centrality rank	Causal centrality rank	Event description
14	10	The camera pans into an animated scene with a teenage girl in a room in a tall building. The camera is inside the apartment, and the girl opens the pizza box and grabs a pizza.
13	9	The girl, before she can eat, hears a knock on the door. The girl opens the door and looks around to see no one is there.
9	7	The girl looks down and sees a package for her in an envelope at her doorstep. The girl goes back to her chair and to her pizza and opens the envelope. The girl pulls out a package with a disc inside it that says "A Single Life."
12	7	The girl pulls out the record disc from the package and the title "A film by: Job, Joris, and Marieke"
4	6	The girl gets up and puts the record disc into the record player and puts down the needle. The song on the disc plays and she sits down and begins to eat her pizza.
6	5	The girl is about to eat the pizza but then there is a flash. She stops and then the song plays its lyrics on the song. Part of the pizza is gone.
3	7	She notices the pizza is eaten and then looks at the disc. The camera pans to the disc playing on the record player, and the disc is spinning on the player.
2	1	The girl stops the disc and there's a record scratch. The girl pulls the disc back and forth on the player and pizza disappears and reappears as she tests it back and forth. The girl pulls the disc forth and the pizza pie disappears completely in the box as well as in her hand. Then she pulls it back and the pizza reappears.
1	6	The girl realizes her power and gets up then lifts the needle on the record player. The disc plays the song and then the flash goes to her as a pregnant woman. The woman stops the record player and stops the song from playing.
10	4	The woman pulls the disc forward and back and sees the baby develop and devolve like the pizza before. The woman pulls the disc forward and the baby develops and pops into her arms but the baby starts crying.
5	4	The woman then flashes into her childhood self and looks at herself. The girl goes up to record player and tries to stop it but pops off the needle.
7	3	The girl flashes to her in a wheelchair as an elderly woman and she looks at herself. The woman rolls up her wheelchair but she flashes back to the same scene over and over again, getting frustrated. The woman rolls up again and flashes but then stops rolling up and finds that nothing happens. She then tries to roll really fast but falls back.
8	2	The woman gets up and she is an old woman who needs a walker and is wearing glasses. The woman sees that the song is about to end and tries to get to the record player.
11	8	The woman turns into ashes in a pot in the same nursery home and the record player stops with the needle lifting.

2.) I was a bit confused about the "onset" analysis reported in Figure 6b. The "offset" analysis (6a) makes sense to me-- e.g., it shows that the hippocampus might be playing a role in

*encoding a just-concluded high-centrality event. But what is the motivation driving the onset version of the analysis? For example, how would the participant "know" that they were about to experience a high-centrality event? Some clarification would be useful. Or alternatively, perhaps there is a different way of getting at the question of whether the hippocampus might play a role in encoding event onsets. One possibility would be to look at the *preceding* event's identity as a high-centrality or low-centrality event. For example, is the "offset effect" (6a) stronger when two high-centrality events occur in succession?*

: The reviewer is correct that at the onset of each event, the participants did not know that they were about to experience a high centrality event or a low centrality event. There is no reason for high and low centrality events to differ before they unfold. Thus, it is expected that no significant difference would be observed between the high and low centrality conditions in terms of their hippocampus responses locked to the onsets of events. This is exactly what we found (Figure 6b). In other words, the onset version of the analysis was the control analysis for the offset analysis. To clarify this, we revised the text in the results section as below:

p.14. In contrast, hippocampal responses following the onset of high vs. low centrality events (**i.e., before the events fully unfold and diverge in terms of their semantic contents**) were not significantly different from each other (Figure 6b), confirming that semantic centrality specifically affected the encoding of information accumulated during just-concluded events.

Minor comments:

1.) The authors define centrality as normalized degree. However, this might conflate some important event information, and I wonder if this could be resulting in lower correlations with centrality. In particular, suppose that two events (A and B) have the same degree, but event A is connected to higher-centrality events than event B. By the current measure, A and B would be considered equally central. But an alternative measure, such as eigenvector centrality, would reflect that A has more "influence" in the network than B.

: We agree with the reviewer that degree centrality might not provide the most complete picture of the connectedness between nodes. Following the reviewer's suggestion, we computed the eigenvector centrality of the nodes in our semantic similarity narrative networks and found that eigenvector centrality and degree centrality (both normalized within each movie) were highly similar to each other ($r = .9991$). This is probably due to the characteristics of our semantic centrality network structure (e.g., each node is connected to every other node, although with different edge weights). Below we show the time courses of degree centrality and eigenvector centrality. Given the high correlation between the measures, we believe that eigenvector centrality would result in the same behavioral and neural effects as the current degree centrality that we used.

2.) The authors might consider describing how centrality was computed in the main text, and possibly even adding a figure with some intuitions about how high vs. low centrality events "look like" in the network (e.g., something like Fig. 1b, but calling out a detail of a high and low centrality event). Although the dot sizes in Figure 1 are useful, it takes quite a bit of zooming in to really appreciate the connections to any single event.

: We apologize for the ambiguity in describing how we computed our centrality metric. The method of computing centrality is described in the Method section (p.25. "The centrality of each individual event within a movie was defined as the degree of each node (i.e., the sum of the weights of all edges connected to the node) in the network, normalized by the sum of degrees and then z-scored within each movie."), but we revised the Results section as below for further clarification in the main text:

p.8. Our main variable of interest reflecting the inter-event narrative structure was the centrality of individual events within a narrative network (Figure 1c). **An event's centrality was computed as its degree (i.e., the sum of the weights of all connections to the event) normalized within each movie.**

In addition, we have clarified our description of the two types of centrality measures in the Introduction:

pp.4-5. In this method, each narrative is transformed into a network of interconnected events based on semantic similarity measured from sentence embedding distances (**the**

“semantic” narrative network). We then calculate *semantic centrality* for each event as the node degree, a graph metric which quantifies the number and strength of connections that a node (event) has to other nodes in the network. Behavioral results revealed that events with higher semantic centrality were more likely to be recalled, without showing primacy and recency effects typical in traditional random list memory experiments^{3,28}. High centrality events were also associated with the neural signatures of stronger and more accurate recall: greater activation and more consistent neural patterns across individuals in the DMN areas including the posterior medial cortex (PMC). The hippocampus showed higher activation following the offset of high centrality events, suggesting that stronger hippocampus-mediated encoding contributed to the high centrality advantages. **In parallel, we created a “causal” narrative network for each movie based on causal relations between events defined by human judgments. Causal centrality of events, again defined as node degree in the network, predicted memory success and neural responses in a similar way to semantic centrality, but also made an independent contribution to each.**

We also added a supplementary figure (Supplementary Figure 3) as below to show the semantic similarity matrices and networks of all ten movies used in the fMRI experiment, highlighting the high centrality nodes (top 40%) in the networks with brighter node colors.

Supplementary Figure 3. Semantic narrative networks of all movie stimuli. **a.** Semantic similarity matrices of the 10 movies used in the fMRI experiment. **b.** Semantic narrative networks of the 10 movies used in the fMRI experiment. Node size is proportional to centrality (normalized degree) computed from unthresholded networks. Edge thickness is proportional to edge weights. Nodes with brighter colors indicate high (i.e., within the top 40% in each movie) semantic centrality events.

Likewise, we revised Supplementary Figure 4b (former Supplementary Figure 3b) and Supplementary Figure 7a-b (former Supplementary Figure 5a-b) as below by highlighting the high centrality nodes with brighter node colors.

Supplementary Figure 4b:

Supplementary Figure 7a-b:

3.) A few of the reported cutoffs for the ISFC seemed arbitrary to me and could benefit from some additional explanation or intuition. For example, the authors first use a 22.5 second (15 TR) cutoff when examining hippocampal-PMC ISFC, and then they report "comparable" results when using a 19.5 second (13 TR) cutoff. Is the analysis sensitive to the precise cutoff? Perhaps it would be cleaner to report results for a *range* of cutoff values? Or to report the range over which the results are "comparable" in some way?

: We apologize for not clearly justifying our choice of the event duration cutoff values. We originally chose 22.5 seconds (15 TRs) based on a prior study (Gonzalez-Castillo et al., 2015). This study showed that functional connectivity patterns measured within time windows as short as 22.5 seconds reliably predicted cognitive states, although 22.5 seconds is much shorter than the time windows used in most functional connectivity studies. Also, note that there is a tradeoff

between the number of time points used to compute functional connectivity and the number of events used to compute the correlation with semantic centrality. That is, because many of our movie events were shorter than 22.5 s (mean event duration = 13.3 s), using longer cutoff values would result in fewer data points (i.e., movie events) and thus less reliable correlation results. At the same time, using shorter cutoff values would result in fewer time points for computing functional connectivity, and again, less reliable results. We had provided the justification for selecting the 22.5 s cutoff in the Method section, but we further revised the Result section as below to clarify this point:

p.15. We first computed ISFC between the hippocampus and PMC during the 26 movie events which were 22.5 seconds (15 TRs) or longer. **Functional connectivity patterns computed within windows as short as 22.5 seconds have previously been shown to robustly predict cognitive states⁴¹.**

In addition, we now report the results using a range of different event duration cutoff values in Supplementary Table 5, as the reviewer suggested. We generally observed similar results regardless of the specific choice of cutoff values (i.e., positive correlations between centrality and hippocampus-PMC ISFC, no significantly positive correlation or even numerically negative correlations between centrality and hippocampus-EVC ISFC). We also revised the text as below:

pp.15-16. We then correlated the ISFC values with the semantic centrality of the events. We found that the hippocampal-PMC interaction was stronger for higher centrality events ($r(26) = .49, p = .01, 95\% \text{ CI} = [.13, .74]$). In contrast, the hippocampal-EVC interaction did not show a significant relationship with centrality ($r(26) = .01, p = .95, 95\% \text{ CI} = [-.38, .4]$), and the correlation was significantly lower than that between hippocampus-PMC ISFC and centrality (95% CI of the difference between correlations⁴² = $[.05, .87]$). **Similar results were observed using different minimum event duration thresholds (Supplementary Table 5).**

Supplementary Table 5. Relationship between semantic centrality and hippocampal-cortical intersubject functional connectivity (ISFC) during movie watching.

Minimum event duration threshold (sec)	Number of events	Correlation between semantic centrality and hippocampus-PMC ISFC (a)			Correlation between semantic centrality and hippocampus-EVC ISFC (b)			95% CI ² of (a) - (b)
		r	p	95% CI ¹	r	p	95% CI ¹	
27	14	.61	.02	[.11, .86]	-.33	.24	[-.73, .24]	[.1, 1.46]
25.5	16	.59	.02	[.14, .84]	-.32	.23	[-.7, .21]	[.12, 1.42]

24	19	.52	.02	[.09, .79]	-.17	.49	[-.58, .31]	[.01, 1.21]
22.5	26	.49	.01	[.13, .74]	.01	.95	[-.38, .4]	[.04, .87]
21	31	.38	.04	[.02, .64]	.02	.93	[-.34, .37]	[-.04, .72]
19.5	44	.29	.06	[-.01, .54]	.05	.75	[-.25, .34]	[-.07, .53]
18	55	.21	.12	[-.05, .45]	-.02	.88	[-.28, .25]	[-.08, .52]

¹ Confidence Interval of the correlation coefficient, [lower bound, upper bound].

² Confidence interval of the difference between two overlapping correlations based on dependent groups, computed using the method described in ref.⁴².

4.) *At several points throughout the manuscript, the way the authors report "null" results (e.g., no significant difference between conditions) might imply that there is evidence of no difference (e.g., lines 165--168, 219--220, 398--400). I'd suggest clarifying that no conclusions can be drawn about those results, rather than that they imply a lack of effect per se.*

: We completely agree with the reviewer that null results do not necessarily imply a lack of effect. We revised the text in the result section accordingly, as listed below:

p.7. Specifically, **we did not find a significant difference** between the mean recall probabilities of the first/middle/last three events of each movie ($F(2,18) = .78, p = .47, \eta^2 = .05$).

pp.9-10. Consistent with the behavioral results from the fMRI experiment, semantic centrality ($\beta = .17, SE = .03, \chi^2(1) = 48.52, p < 0.001$) and causal centrality ($\beta = .44, SE = .03, \chi^2(1) = 255.67, p < 0.001$) each uniquely predicted the successful recall of an event, without any clear evidence of serial position effects (i.e., **no statistically significant difference** between the mean recall probabilities of the first/middle/last three events of each movie, $F(2,18) = .85, p = .44, \eta^2 = .04$).

pp.15-16. In contrast, the hippocampal-EVC interaction **did not show a significant relationship with centrality** ($r(26) = .01, p = .95, 95\% CI = [-.38, .4]$), and the correlation was significantly lower than that between hippocampus-PMC ISFC and centrality (95% CI of the difference between correlations⁴² = $[.05, .87]$). Similar results were observed using different minimum event duration thresholds (Supplementary Table 5).

5.) I'd be curious as to how the authors settled on using USE as opposed to other available models. Was the decision somewhat arbitrary (or based on ease of implementation), or was there a deeper theoretical reason for embedding text using USE?

: We chose to use the Universal Sentence Encoder because it was considered the most advanced and effective sentence embedding method that had just been introduced around the time we started the current project (summer 2018). Since our movie annotations did not provide enough amount of text data for training other types of models such as topic modeling, we had to find pre-trained natural language models, and USE was the one that performed the best in various natural language processing tasks (e.g., sentence similarity) compared to other sentence embedding methods available at the moment. Another possible option was to use the average of word2vec embeddings of the individual content words within the sentences describing each movie event. This method produced results highly similar to those obtained using USE: the event-wise semantic centrality based on USE vectors and the centrality based on the word2vec vectors (averaged across all content words in each event description that exist in the word2vec database) were significantly correlated with each other ($r = .54, p < .001$). In addition, the word2vec-based semantic centrality was correlated with recall probability ($r = .34, p < .001$), as the USE-based semantic centrality did.

Thus, we found that the two approaches are both valid ways of defining semantic similarity between events. However, we chose to use USE over the word2vec-averaging method for its simplicity, because USE takes full sentences as input and thus does not require additional steps and computations such as selecting content words and averaging embeddings. This also allowed us to minimize the preprocessing of the annotation texts and preserve the natural descriptions of the human annotators as much as possible.

We have added the results of the above analyses to the text, also in response to Reviewer 2's Conceptual Comment C:

p.25. *Semantic narrative networks.* Movie annotations were used to generate narrative networks based on the semantic similarity between events (Figure 1). For each annotator and movie, the text descriptions for the fine-grained sub-events were concatenated within each movie event. The text descriptions were then encoded into high-dimensional vectors with Google's Universal Sentence Encoder (USE²⁵) such that each movie event was represented as a 512-dimensional vector. The USE vectors from the three annotators were highly similar to each other (mean event-wise cross-annotator cosine similarity between all possible annotator pairs = .78; Supplementary Figure 1); thus the USE vectors were averaged across annotators within each movie event. For each movie, the narrative network was generated by using the cosine similarity between the USE vectors of movie event pairs as the edge weights between nodes (events). **The semantic centrality values based on USE sentence embedding vectors were correlated with those based on word-level overlap or word2vec embeddings (Supplementary Methods).**

Supplementary Methods pp.1-2. *Semantic narrative networks based on word-level information.*

To test the effects of semantic centrality based on word-level rather than sentence-level similarity between movie event annotations, we created two additional types of semantic narrative networks. First, we created narrative networks whose edge weights between events were defined as Jaccard indices reflecting the word overlap (exact matching words) between event text descriptions. The Jaccard indices were computed within each annotator, and then averaged across annotators within each movie. As in the USE-based narrative networks, event centrality was defined as the normalized node degree. We found that the semantic centrality computed from the networks based on Jaccard indices was positively correlated with the semantic centrality based on USE embeddings ($r(202) = .64, p < .001, 95\% \text{ CI} = [.55, .71]$) and also with recall probability ($r(202) = .27, p < .001, 95\% \text{ CI} = [.14, .39]$). Second, we created networks whose edge weights between events were the cosine similarity between the word embeddings of the events. Specifically, the word embedding of an event was generated by averaging the word vectors (based on Google's pre-trained Word2Vec model; GoogleNews-vectors-negative300-SLIM) of unique words contained in the text description of the event, separately for each annotator. The word embeddings were then averaged across annotators. Words that were not included in the Google's database were excluded from the analysis. The centrality (normalized node degree) computed from these networks was again positively correlated with the USE-based semantic centrality ($r(202) = .54, p < .001, 95\% \text{ CI} = [.43, .63]$) and with recall probability ($r(202) = .34, p < .001, 95\% \text{ CI} = [.21, .46]$).

REVIEWER COMMENTS

Reviewer #1 (Remarks to the Author):

The reviewers have satisfactorily addressed my original comments. The other reviewers raised some excellent points and I think that the authors have done a thorough job at addressing these too. In my opinion this is a very good paper and will make a very useful contribution to the field.

Reviewer #3 (Remarks to the Author):

The reviewers have done a commendable and thorough job with their revision of this paper. I have no further concerns.